# Fairness for the People, by the People: Minority Collective Action

## Abstract

Machine learning models often preserve biases present in training data, leading to unfair treatment of certain minority groups. Despite an array of existing firm-side bias mitigation techniques, they typically incur utility costs and require organizational buy-in. Recognizing that many models rely on user-contributed data, end-users can induce fairness through the framework of Algorithmic Collective Action, where a coordinated minority group strategically relabels its own data to enhance fairness, without altering the firm's training process. We propose three practical, model-agnostic methods to approximate ideal relabeling and validate them on real-world datasets. Our findings show that a subgroup of the minority can substantially reduce unfairness with a small impact on the overall prediction error.

## 1 Introduction

As machine learning (ML) tools become increasingly accessible, more firms deploy them for decision-making. However, ML models often perpetuate societal biases present in their training data, leading to unfair outcomes across demographic groups (Barocas & Selbst, 2016). Moreover, most fairness-preserving learning algorithms incur a non-negligible cost in accuracy or computational resources (Menon & Williamson, 2018; Zhao & Gordon, 2019; Dehdashtian et al., 2024; Sadeghi et al., 2022), which can discourage practical adoption.

Since firms control the ML pipeline, end-users lack access to directly enforce fair treatment. Yet, affected users routinely generate and share data, through clicks, ratings, or other contributions, that is used to train the firm's models. Such cases, where users collaborate to influence what firms learn, are not uncommon and are well-documented (Sigg et al., 2025). Consequently, if underrepresented minority groups collaboratively alter the data they share, they might be able to steer the learned model towards fairer behavior, even without access to the firm's training pipeline.

For example, consider a human resources company that makes a profit by filling vacancies and trains ML models on resumes to predict the skills of the candidates. While the majority may have more formal education and college degrees, disadvantaged groups may have informal training or internships. As a result, the ML model does not assign the correct skills to the minority due to their lack of formal education, despite their practical experience. The minority members can react to this injustice by collectively submitting their resumes, but reframing their reported skills, such as sales or management. Appendix A describes other examples where such *collective action* is applicable.

This idea is reminiscent of *pre-processing* fairness techniques (Kamiran & Calders, 2009; Luong et al., 2011; Zemel et al., 2013; Madras et al., 2018), which modify the data before model training. Unlike these prior approaches, which assume centralized control over the data, we consider the setting of *algorithmic collective action* (ACA) (Hardt et al., 2023; Ben-Dov et al., 2024; Baumann & Mendler-Dünner, 2024; Sigg et al., 2025; Gauthier et al., 2025), in which a small group of users strategically modifies their own data to influence the correlations learned by the model.

We adapt the *erasure strategy* from Hardt et al. (2023) to reduce predictive correlation between group membership and the target label by relabeling minority samples. The collective is restricted to members of the minority group since minority members are more motivated to join collective action (Saleem et al., 2021; Begeny et al., 2022) and can be efficiently mobilized (McAdam, 1999; Michelson, 2005), while majority-group users may be less inclined to disrupt the status quo. We show that when a classifier is trained on data affected by this form of ACA, standard fairness metrics

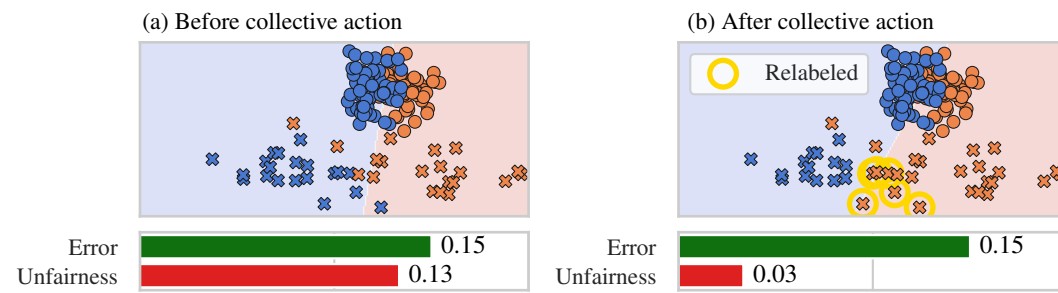

Figure 1: Minority-only collective action can substantially improve fairness. With only 6 label flips, the fairness violation of logistic regression goes down by over 75% with only a negligible increase in prediction error. Circles and crosses represent majority and minority points, respectively.

improve substantially. This improvement is illustrated in Figure 1, where a small minority collective significantly reduces unfairness with minimal impact on prediction error.

The key obstacle in implementing the erasure strategy is that it requires knowledge of each user's label under a counterfactual group membership. Computing such counterfactual labels exactly would require access to an underlying causal model, which is typically infeasible in practice. To overcome this challenge, we propose three *model-agnostic* methods to estimate the counterfactual labels.

To summarize, our main contributions are: **(1)** We divert from the common firm-side fairness methods and focus on user-side by introducing the setting of **minority-only algorithmic collective action for fairness** in ML (Section 2), and design three algorithms that Pareto-dominate the random-choice baseline. **(2)** Through experiments on benchmark datasets, we demonstrate that these algorithms can **significantly improve fairness metrics** with only a slight accuracy cost and few label flips, and minimal knowledge of majority-group data. **(3)** We investigate **fundamental limitations of minority-only collectives** and provide theoretical results showing that **better representations and better counterfactual approximation methods** can improve these algorithms.

## 2 COLLECTIVE ACTION FOR FAIRNESS

To establish the connection between fairness and ACA, Section 2.1 first defines the problem setting and how unfairness can be measured. Then, Section 2.2 describes the theoretical framework of ACA and how it can be utilized to mitigate bias. Finally, Section 2.3 formally relates between ACA to group fairness metrics through counterfactual fairness.

### 2.1 GROUP FAIRNESS FOR CLASSIFICATION

We consider a setting in which a firm uses ML to predict a binary label $y \in \{0, 1\}$. The firm collects data from its users, forming a dataset $\mathcal{D} = \{(x_i, a_i, y_i)\}_{i=1}^{n}$, where $x_i \in \mathbb{R}^m$ denotes user $i$'s feature vector, $a_i \in \{0, 1\}$ is a sensitive attribute indicating binary group membership ($a_i = 0$ for the majority group, $a_i = 1$ for the minority), and $y_i \in \{0, 1\}$ is the true label. We assume the users are drawn independently and identically distributed (i.i.d.) from a distribution $\mathbb{P}_0$ over $\mathbb{R}^m \times \{0, 1\} \times \{0, 1\}$. The firm trains a classifier $h : \mathbb{R}^m \to \{0, 1\}$ to minimize the prediction error, defined as

$$\text{Error}(h) = \mathbb{P}[h(x) \neq y]. \tag{1}$$

To do so, the firm minimizes the empirical error on $\mathcal{D}$ via Empirical Risk Minimization (ERM).

In the group-fairness paradigm, the sensitive attribute $a \in \{0, 1\}$ partitions the data into subgroups, and fairness criteria seek to ensure similar outcomes across these groups. Common metrics include statistical parity (SP) (Calders et al., 2009; Dwork et al., 2012) and equalized odds (EqOd) (Hardt et al., 2016). In this work, we focus primarily on violations of EqOd, formally defined as

$$\text{EqOd}(h) = \frac{1}{2} \sum_{z=0,1} |\mathbb{P}[h(x) = 1 | a = 1, y = z] - \mathbb{P}[h(x) = 1 | a = 0, y = z]|, \tag{2}$$

which measures the differences between true positive and false positive rates. Appendix B.1 provides formal definitions and further discussion of these metrics.

ERM-trained models tend to achieve low predictive error, but this often comes at the cost of fairness violations under SP and EqOd (Menon & Williamson, 2018; Zhao & Gordon, 2019; Bardenhagen et al., 2021; Sanyal et al., 2022). Despite significant progress in fairness research, most solutions have traditionally focused on *firm-side* solutions: pre-processing the dataset, in-processing modifications to the training algorithm, or post-processing the classifier's predictions. These approaches almost always incur errors or additional pipeline complexity, discouraging firms to deploy them in practice.

While most prior work has focused on firm-side solutions, this work shifts the focus to *user-side* methods that do not require the firm's participation. Since users generate the training data, they can collectively influence the learned model by strategically modifying their own behavior. For instance, consider a digital platform that recommends content to a user based on classifier predicting engagement labels $y_i \in \{$will engage, will not engage$\}$. The classifier, trained on historical user interactions, may unintentionally rely on group membership rather than individual preferences when making recommendations for minority members. In response, users can coordinate to alter their interaction patterns, such as clicking on or avoiding certain items. This ACA affects the dataset in a way that steers the learned classifier toward fairer outcomes, and is generally studied under the field of algorithmic collective action (Hardt et al., 2023).

## 2.2 Algorithmic collective action

In social sciences, *collective action* refers to the coordinated efforts of individuals working together to pursue a shared goal (Olson, 1989; Marwell & Oliver, 1993). Hardt et al. (2023) adapt this notion to ML, proposing that a group of users, termed a collective, can strategically modify their data to align the behavior of a trained classifier $h$ with the collective's goals. In this formulation, the training distribution is a mixture distribution $\mathcal{D} \sim \mathbb{P}_\alpha = \alpha \mathbb{P}^* + (1-\alpha)\mathbb{P}_0$, where $\mathbb{P}^*$ and $\mathbb{P}_0$ are the collective and base distributions, and $\alpha \in [0,1]$ denotes the proportion of the collective.

**Relation to fair representation learning.** With user agency over the data, one possible form of ACA for fairness is to modify their features to increase correlation with the label $y = 1$. An analogous firm-side approach is fair representation learning (FRL), which learns a transformation from the input space to a representation space such that ERM leads to a classifier that is both accurate and fair (Zemel et al., 2013; Jovanović et al., 2023). However, a hindrance of FRL in the context of ACA is that the transformation must be applied consistently at inference time, requiring active cooperation from each minority member to transform their features. In contrast, our setting assumes users have control only over the labels and cannot intervene in other parts of the machine learning pipeline.

**Erasing a signal.** Suppose the collective seeks a classifier that is invariant under a transformation $g : \mathbb{R}^m \to \mathbb{R}^m$ applied to the features. The success of the collective can be quantified as

$$S(\alpha) = \mathbb{P}_0 \left[ h(g(x)) = h(x) \right], \tag{3}$$

the probability, under the base distribution, that the classifier's prediction remains unchanged after applying $g$ to the features. In words, the collective's goal is to *erase the signal* $g$: to ensure the classifier behaves identically regardless if the $g$ is applied. Intuitively, if $g$ embeds a feature pattern correlated with group membership (i.e., minority or majority), then achieving invariance under $g$ promotes fairness by reducing the classifier's dependence on group-identifying information.

To achieve signal erasure, Hardt et al. (2023) propose the collective relabels itself with the most likely label under the transformation $g$. Formally, the strategy is defined as

$$x, y \to x, \underset{y' \in \{0,1\}}{\operatorname{argmax}} \mathbb{P}_0 \left( y' | g(x) \right). \tag{4}$$

Since this strategy leaves the features unchanged, it is well-suited for settings where the minority is limited to modify only their labels, such as ours. For $\epsilon$-optimal Bayes classifiers (Definition 2 in Appendix B.2), Hardt et al. (2023) prove the following lower bound for its success

$$S(\alpha) \geq 1 - \frac{2(1-\alpha)}{\alpha} \cdot \tau - \frac{\epsilon}{(1-\epsilon)\alpha}, \tag{5}$$

where $\tau = \underset{x \sim \mathbb{P}_0}{\mathbb{E}} \left[ \underset{y' \in \{0,1\}}{\max} |\mathbb{P}_0(y'|x) - \mathbb{P}_0(y'|g(x))| \right]$ captures the sensitivity of $y$ under $g$.

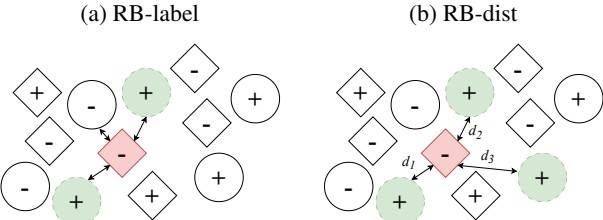

Figure 2: Visualization of KNN scoring methods with $k=3$. The minority is represented by the squares and the majority by circles, marked with a positive "+" or a negative "−" label. (**a**) *RB-label*: Two of the nearest majority neighbors have a positive label, resulting in the score $s=2$. (**b**) *RB-dist:* The average distance to the nearest positive majority neighbors results in the score $s=-\left(d_1+d_2+d_3\right)/3$.

Note that the strategy in Equation (4) may require some majority members to relabel themselves with the label $y = 0$. Such a change might deter them from participating in the collective action, either because majority members are unwilling to give up their advantage or prefer to maintain the status quo. To avoid this conflict, we restrict the collective to include only minority members. We discuss the implications of this restriction in Section 5.

### 2.3 COUNTERFACTUAL FAIRNESS

The concept of *counterfactual fairness* (CF) (Kusner et al., 2017; Garg et al., 2019; Wu et al., 2019) bridges between signal erasure success to group fairness. To introduce this idea, assume that a sample $x$ is generated by a causal model, in which the group membership $A$ is a causal parent. Then a classifier $h$ is counterfactually fair if its predictions are invariant to interventions on the group membership, i.e., $h(x) = h(x_{A \leftarrow a'})$ for any $a'$, where $x_{U \leftarrow u}$ denotes an intervention on a causal parent $U$ of a sample $x$. In certain causal contexts, CF implies or aligns with group fairness criteria such as SP or EqOd (Anthis & Veitch, 2023). Therefore, if ACA induces a counterfactually fair classifier, it may also induce a fair classifier under SP or EqOd.

As focus is on fairness for the minority, we relax the original definition of CF (Kusner et al., 2017).

**Definition 1.** *A classifier $h$ is **minority-focused counterfactually fair** if under any context $X = x$,*

$$\mathbb{P}_0\left(h\left(x_{A\leftarrow a}\right) = y | X = x, A = 1\right) = \mathbb{P}_0\left(h\left(x_{A\leftarrow a'}\right) = y | X = x, A = 1\right), \quad (6)$$

*for any value $a'$ attainable by $A$.*

By this definitions, changing the group membership of a minority individual, in a counterfactual sense, has no effect on the classifier's prediction. ACA can theoretically enforce such fairness by applying the erasure strategy from Equation (4) with the counterfactual signal $g(x) = x_{A \leftarrow 0}$, which replaces a minority individual with its majority-group counterfactual. This ACA aligns the signal erasure success from Equation (3) with minority-focused counterfactual fairness from Definition 1. The following proposition, proved in Appendix C.1, formalizes this alignment.

**Proposition 1.** *A Bayes classifier trained on $\mathbb{P}_\alpha$ is minority-focused counterfactually fair if and only if the success of a minority collective is $S = 1$.*

This result directly connects between ACA theory to fairness. Thus, perfect success of the collective is equivalent to achieving minority-focused counterfactual fairness.

## 3 APPROXIMATING THE COUNTERFACTUAL LABEL

This section describes how a minority collective can approximate a signal-erasure strategy to promote fairness in practice. While the theory of signal erasure has been studied before (Hardt et al., 2023; Gauthier et al., 2025), prior work lacks empirical evaluation. In this paper, we present the first practical algorithm for signal erasure and provide experimental results in Section 4. As discussed in Section 2.3, a suitable signal to erase is $g(x) = x_{A \leftarrow 0}$, where each collective member relabels themselves according to Equation (4).

However, end-users lack access to the true causal model and cannot compute the counterfactual labels directly. To address this limitation, we propose to assign each collective member $i$ a score $s_i$,

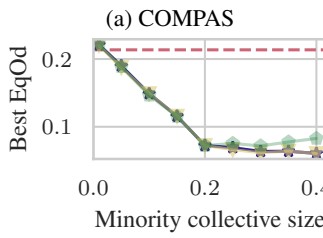 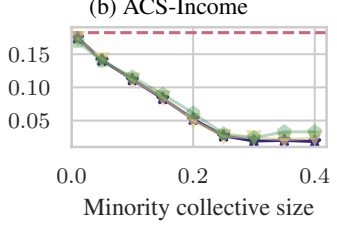 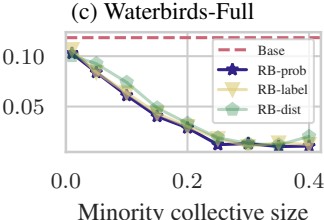

Figure 3: The lowest EqOd violation a collective can achieve greatly improves as the collective size increases, up to a certain point. Each point is a mean of 10 runs, with the standard deviation being smaller than the markers. In all the datasets we experimented on, the lowest EqOd violation converges around $\alpha = 0.3$. Additional results are presented in Figure 11 in the appendix.

which serves as a proxy for the likelihood that they would receive the label $y = 1$ if they belonged to the majority. Given a budget of $M$ label flips, the collective selects the $M$ members with the highest scores; these individuals flip their labels from $y = 0$ to $y = 1$. The budget $M$ controls the accuracy–fairness tradeoff, where a higher budget typically leads to better fairness, but higher error.

We introduce three scoring functions, each capturing a different notion of similarity to majority users:

1. **Rank by probability (RB-prob):** Train a regressor $f : \mathbb{R}^m \to \mathbb{R}$ on exclusively majority data ($a = 0$) to estimate the probability $\mathbb{P}(Y = 1 | X = x)$ of having the label $y = 1$. Each collective member $i$ receives a score based on the model's prediction:

$$s_i = f(x_i). \tag{7}$$

2. **Rank by label (RB-label):** For each collective member $i$, identify the set $K_i$ of their $k$ nearest majority neighbors using Euclidean distance. The score is the number of neighbors with the label $y = 1$:

$$s_i = \sum_{j \in K_i} \mathbf{1}\{y_j = 1\}. \tag{8}$$

3. **Rank by distance (RB-dist):** Restrict the neighbors set $K_i$ to only majority users with the label $Y = 1$. The score is the negative mean Euclidean distance to these neighbors:

$$s_i = -\frac{1}{k} \sum_{j \in K_i} \|x_i - x_j\|_2. \tag{9}$$

Intuitively, RB-prob assigns a higher score where a classifier trained solely on majority data predicts a higher likelihood of the label $y = 1$. RB-label scores collective members according to the frequency of $y = 1$ among their majority neighbors, while RB-dist prioritizes those who are closer majority users with $y = 1$. Figure 2 provides visualizations for RB-label and RB-dist.

## 4 EXPERIMENTAL RESULTS

This section evaluates the performance of our methods. We compare the three methods, RB-label, RB-dist, RB-prob, against a random baseline that flips $y = 0$ labels to $y = 1$ for $M$ randomly selected collective members. We conducted experiments on the tabular datasets COMPAS (Mattu et al., 2016), Adult (Becker & Kohavi, 1996), HSLS (Jeong et al., 2022), ACS-Income (Ding et al., 2021), the image dataset Waterbirds (Sagawa et al., 2020) and the text dataset CivilComments (Borkan et al., 2019). For Waterbirds, we use features extracted from a pre-trained *ResNet-18* (denoted Waterbirds-Full) and for CivilComments, we used the extracted features from Hugging Face's pre-trained *bert-base-uncased* model (denoted CivilComments-Full). In addition to the complete features of Waterbirds and CivilComments, we also include experiments on the PCA features, with 85 components for Waterbirds (denoted Waterbirds-PCA) and 100 components for CivilComments (denoted CivilComments-PCA). Details on the datasets and the pre-processing are provided in Appendix D.1.

All reported metrics are computed on a fixed test set, without any ACA, and averaged over 10 independent runs for each method described in Section 3. In each run, we randomly selected a minority collective to apply the method. For the KNN-based methods, we tuned the neighborhood

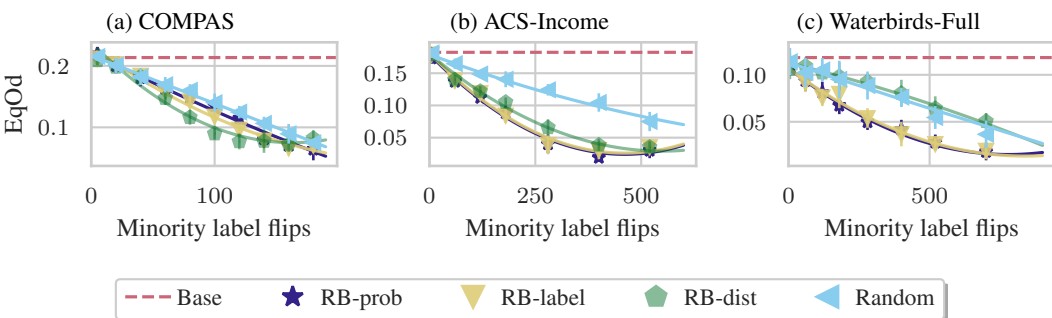

Figure 4: Our proposed methods are consistently more efficient than randomly flipping labels, requiring less label flips to attain the same level of EqOd. Each marker is the mean of 10 random runs with a specific number of label flips. The standard deviation is presented by the error bars. The dashed line shows the mean EqOd for a classifier trained on the dataset without collective action.

size $k$ using a $15\%$ validation split from the train set, optimizing for EqOd and SP. Finally, we trained a gradient-boosted decision tree on each modified train set. More technical details are given at Appendix D.2 and the complete set of results can be found in Appendix E.2.

**Importance of collective size**    While the number of label flips $M$ is the primary factor for balancing between accuracy and fairness, the size of the collective, $\alpha$, also plays a role. In addition to bounding the possible number of flips, increasing $\alpha$ also expands the candidate pool from which the most effective labels to flip can be selected. To measure this effect, the experiments included a range of $\alpha$ values, each tested with multiple values of $M$. For each $\alpha$, we define the best achievable EqOd as the minimum EqOd across all tested values of $M$. As shown in Figure 3, increasing $\alpha$ improves the best achievable EqOd until saturating around $\alpha = 0.3$. We fix this value for all remaining experiments.

**Flipping cost**    Since each method scores candidates differently, they may also vary in efficiency, that is, the number of label flips required to achieve a given level of fairness. To evaluate efficiency, Figure 4 plots EqOd as a function of number of label flips $M$, where lower curves indicate more efficient methods. The random baseline consistently yields the worst EqOd across all values of $M$, highlighting the value of informed relabeling algorithms. However, no single method dominates the others in all settings: While RB-prob and RB-label often outperform the other methods, RB-dist can surpass them in specific cases (e.g., Figure 4a), or perform comparably to the random baseline in others (Figure 4c). These results suggest that a well-chosen scoring function enables the collective to achieve a desired level of fairness with fewer label flips, reducing the "cost" of ACA and mitigating the accuracy loss from excessive relabeling.

Interestingly, beyond a certain number of flips, EqOd begins to increase, indicating that excessive flipping can shift unfairness from the minority to the majority. This upturn reflects the fundamental limits of minority ACA for fairness, a point we elaborate on in Section 5.

**Partial knowledge of the majority**    In all previous experiments, we assumed that the collective has full access to the majority data to estimate the counterfactual labels. Here we investigate the performance of our methods when limiting this knowledge. To visualize the fairness-error tradeoff, we measure the error and EqOd for a range of label flips, yielding a set of pairs (Error, EqOd). This set forms a Pareto front, representing the tradeoff. A Pareto front is said to *dominate* another if it lies entirely to the left (lower error) and below (lower unfairness) of the other. The Pareto fronts in Figure 5 exhibit that a collective employing RB-prob, when restricted to only 10 majority members, performs similarly as a collective with full knowledge. While the Pareto fronts remain similar, limited majority knowledge can increase the number of required flips. This is evident when comparing to the zero-knowledge scenario, designated as random in Figure 4. This finding implies that the fewer flips the collective is allowed, the more important it is to have access to the majority data.

## 5    LIMITATIONS OF MINORITY COLLECTIVE ACTION

Previous work on ACA assumes that the collective is uniformly sampled from the distribution $\mathbb{P}_0$ and that the collective has a perfect oracle for the conditional distribution $\mathbb{P}_0 (Y|X)$. Yet, our method

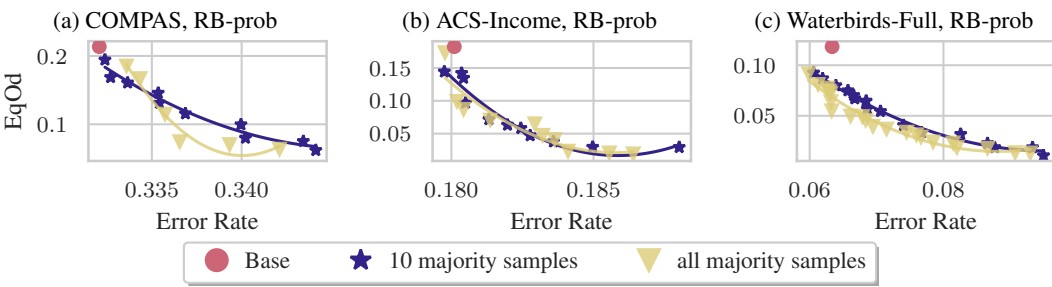

Figure 5: Limiting the knowledge of the collective about the majority does not significantly harm the Pareto front. Each point is the mean of 10 runs and the curves are fitted to guide the eye.

restricts collective participation to minority members and approximates this conditional distribution. Those differences introduce limitations to the existing theory, which we analyze in this section.

**Collective restricted to the minority.** As mentioned above, we focus on collectives composed solely of minority members, unlike prior work. This restriction expresses scenarios in which majority members lack incentives to support changes that would benefit the minority, and instead prefer to preserve the status quo. Naturally, this limitation reduces the collective's impact. Consider a binary classification task on the two-dimensional 4-Gaussian mixture model $\mathbb{P}_{4GMM}$ where each Gaussian belongs to a distinct combination of label and group membership, as illustrated in Figure 6. Each label consists of a large majority subgroup and a significantly smaller minority subgroup. We can then state the following informal result about the EqOd fairness violation of ERM.

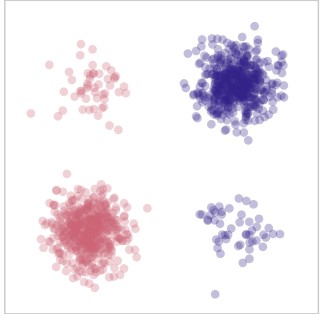

Figure 6: The distribution $\mathbb{P}_{4GMM}$ used in Proposition 2. The color signifies the label, and the density shows the group membership.

**Proposition 2** (Informal)**.** *Consider the dataset $\mathbb{P}_{4GMM}$, where every minority point participates in the ACA by flipping all $y = 0$ labels to $y = 1$. Then, under sufficiently separable clusters, with high probability, the EqOd of the ERM classifier minimizing the logistic loss will asymptotically approach $0.5$.*

The formal Proposition 5 is provided in Appendix C.2 along with all necessary assumptions, which holds for a broader family of distributions and can be extended to any dimensionality $\mathbb{R}^d$ using techniques similar to those in Chaudhuri et al. (2023). Although Proposition 2 is not a formal lower bound, it emphasizes an important limitation: ACA restricted to the minority cannot generally achieve perfect fairness, even under very advantageous conditions involving a maximum-sized collective, a strong strategy, and a disregard for accuracy. This limitation stands in contrast to standard firm-side bias mitigation methods, which in principle, achieve perfect fairness. There are several reasons why relabeling alone may not be enough to get perfect fairness. For one, relabeling according to the counterfactual implicitly assumes that the label is determined by the same features across the majority and minority, but this assumption is not valid under certain distribution shift between the groups. To illustrate, consider a firm training a classifier to screen candidates for a managerial position. Majority members may be more educated, while minority members may have more hands-on experience rather than formal education. In this case, a counterfactual label associated with the majority is disjointed from the features associated with the features, rendering the signal erasure strategy irrelevant. Future work could study this problem and determine when it is beneficial to change the features as well.

We empirically corroborate the findings of Proposition 2 on real world datasets by examining the fairness–accuracy tradeoff of several fair learning methods. Figure 7 compares the Pareto fronts of RB-prob, one of our minority ACA methods, with established firm-side methods. We observe that the lowest fairness violation achievable by RB-prob is greater than that of the firm-side approaches. However, the firm-side methods are able to arrive at perfect fairness only at a cost of prohibitively high prediction error. But, inspecting the region where the error is small compared to the base classifier, the fairness of RB-prob is comparable to that of the firm-side methods.

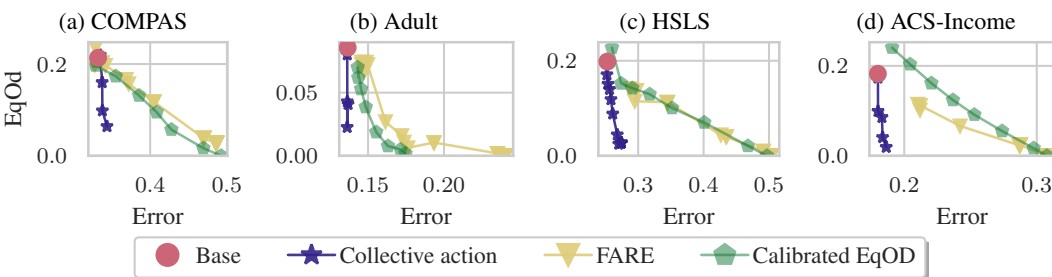

Figure 7: User-side method cannot achieve perfect fairness, while the firm-side pre-processing method FARE Jovanović et al. (2023) and the post-processing method calibrated equalized odds Pleiss et al. (2017) attain 0 EqOd with large error. However, RB-prob's fairness is better than the base classifier, with a smaller error than the firm-side methods.

**Estimating counterfactuals.** In Section 3 we proposed methods to estimate which individuals would receive a counterfactual label that is different than their original label. However, the success lower bound in Equation (5) assumes perfect knowledge of $\mathbb{P}_0$ and of the underlying causal model. To account for the estimation error, we model the collective's prediction as the output of an algorithm $\mathcal{A}(x) \approx \mathbb{P}_0 \max_y (y|x_{A \leftarrow 0})$ that has an error rate $\rho$, defined as

$$\rho := \mathbb{P}_0(\mathcal{A}(x) \neq \arg\max_{y'} \mathbb{P}_0[y'|g(x)]). \tag{10}$$

Given this definition, we derive the following lower bound on success, proved in Appendix C.3.

**Proposition 3.** *With algorithm $\mathcal{A}(x)$ with label error $\rho$, the success of the collective is bounded by*

$$S(\alpha) \geq 1 - \frac{2(1-\alpha)}{(1-2\rho)\alpha}\tau - \frac{\epsilon}{(1-\epsilon)(1-2\rho)\alpha}. \tag{11}$$

This bound recovers Equation (5) when $\rho = 0$, but higher values of the error $\rho$ worsen the bound. Next, we show how to use FRL to reduce the error $\rho$, thereby improving the lower bound.

**Impact of feature representations** Since the methods RB-label and RB-dist rely on KNN, their performance is sensitive to the choice of distance metric and feature representation. In our main experiments, we used Euclidean distance in the original feature space, which is convenient but could be suboptimal. Here, we explore whether FRL can learn a more suitable representation space for KNN. A *fair representation* maps the data into a space where the group-based bias is removed while preserving informative features. Intuitively, such representations may help RB-label and RB-dist to better estimate the counterfactual labels. To formalize this intuition, we consider predicting the counterfactual label of minority points using a 1-NN classifier on majority data, i.e., assigning each minority point the label of its nearest neighbor in the majority. In settings where the minority is distributed differently than the majority (e.g., $\mathbb{P}_{4GMM}$), this task can be challenging. The following informal result compares the error of 1-NN in the original features space to its error in FRL.

**Proposition 4** (Informal). *Let data be drawn from $\mathbb{P}_{4GMM}$, and $\rho_{plain}$ denote the error of a 1-NN classifier that assigns the label of the nearest majority neighbor in the original feature space. Then there exists a fair representation in which a 1-NN classifier achieves error $\rho_{FRL}$ such that, asymptotically with respect to the dataset size, $\rho_{FRL} \leq \rho_{plain}$.*

The formal statement, Theorem 1 can be found in Appendix C.4. The result suggests that FRL can reduce the counterfactual label error $\rho$ of RB-label and RB-dist, consequently improving the lower bound of the collective's success according to Proposition 3. Empirically, Figure 8 indicates that applying FARE (Jovanović et al., 2023) before the KNN step improves the Pareto front for RB-dist. On the other hand, methods that rely purely on predictive information, such as RB-prob, can perform worse, due to FRL inadvertently removing features predictive of the class label.

## 6 RELATED WORK

Optimizing for fairness often comes at the cost of reduced classification accuracy, leading to the well-documented accuracy–fairness tradeoff (Menon & Williamson, 2018; Zhao & Gordon, 2019;

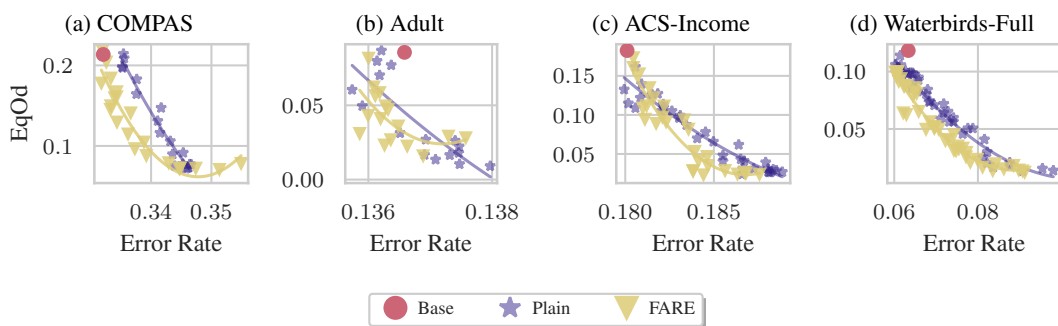

Figure 8: The Pareto fronts for using a fair representation when computing the KNN for RB-dist dominate the Pareto fronts for KNN computed on untransformed features. The blue stars represent the KNN without transforming the data, and the yellow triangles represent the KNN when the data is transformed using FARE Jovanović et al. (2023). The lines are fitted to guide the eye.

Dehdashtian et al., 2024; Sadeghi et al., 2022). In response, previous work has proposed fairness interventions at different stages of the ML pipeline: pre-processing methods modify the training data before learning (Kamiran & Calders, 2009; Luong et al., 2011; Zemel et al., 2013; Jovanović et al., 2023), in-processing methods adjust the learning algorithm itself (Agarwal et al., 2018; Nam et al., 2020; Sagawa et al., 2020; Liu et al., 2021), and post-processing methods correct the predictions of a trained (unfair) classifier (Hardt et al., 2016; Alghamdi et al., 2022; Tifrea et al., 2024; Cruz & Hardt, 2024). A firm can introduce any of these categories into its pipeline, while users, who control only their data can only partially implement pre-processing methods. However, as mentioned in Section 2.2, using feature-changing pre-processing methods such fair representation learning (Zemel et al., 2013; Jovanović et al., 2023) demand changing those features during inference time as well.

Still, some pre-processing methods change only the labels, similarly to our proposed collective action. The method by Luong et al. (2011) compares between the minority KNN and majority KNN and flip the labels according to the difference of positive labels between the two groups of neighbors. This method resembles RB-label, with the difference that RB-label examines only the majority KNN in order to approximate the counterfactual. The approach of Kamiran & Calders (2009) trains a regressor to predict $y = 1$ outcome probabilities, and flip the label of minority members with $y = 0$ labels and high probability according to the regressor to have $y = 1$, and similarly flip majority $y = 1$ labels to $y = 0$. Flipping from both groups supposedly preserves the error of the classifier. Our method RB-prob differs by training the regressor only on the majority to better approximate the counterfactuals. Since this approach requires flipping the labels of majority members as well, it cannot be completey adopted by the collective. In Appendix E.1 we compare the between RB-prob to CND and KDP, and find that our method is more efficient in terms of number of label flips.

## 7 CONCLUSION

This work demonstrates that user-side methods can effectively reduce unfairness in machine learning. While much of the existing fairness research focused on firm-side methods, these often come at a cost that may not be worth to the firm. This catch emphasizes the importance of studying user-side approaches for bias mitigation. We show empirically that ACA can considerably reduce unfairness in a variety of datasets, though not completely. Importantly, we also examine the limitations of a minority being composed of only minority members, and how the success is affected by approximating the counterfactual labels. Our proposed methods require the collective to relabel themselves, which often comes with a price, as the users have to go against their true nature. However, as has been studied on real world cases, minority members willingly participate in collective action to benefit their demographic, after being encouraged by their community (Begeny et al., 2022) or by the media (Saleem et al., 2021) and efficiently mobilized in large-enough scales (McAdam, 1999; Michelson, 2005).

We also note that in general, ACA methods can be exploited by malicious parties seeking self-gain or harming other communities, and it is important to discuss these limitations and possibly regulate them. Overall, this paper shows a practical use case of collective action in the hopes of sparking further research into applications of ACA and user-side methods for social good.

## REPRODUCIBILITY STATEMENT

We publish all code necessary to reproduce the experiments in the paper. The details of how we pre-process he datasets are described in Appendix D.1 and also contained in our codebase. Finally, we provide the training details in Appendix D.2.

## LLM USAGE STATEMENT

We used ChatGPT to rephrase some sentences and find better words to improve the reading flow of the paper.

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

## 8 APPENDIX

## A MOTIVATING EXAMPLES

To recognize real-world problems that lend themselves well to collective action for fairness one needs to look for the following few characteristics:

- Firm and goal: A firm trains a predictive model primarily to minimize average error, with little incentive to protect minority groups.
- End-users: People who use the platform and whose behavior generates data for the firm's dataset.
- Disadvantaged group: A subgroup of end-users who is treated unfairly.
- Relabeling possibility: How the minority can relabel themselves to make the trained classifier fairer.

Here we suggest several cases that answer these characteristics.

1. Content moderation
   - Firm and goal: A global social-media company optimizes a high-recall harmful-content detector measured on its largest user pools.
   - End-users: Everyday users of the platform who can flag offensive content.
   - Disadvantaged group: Slurs, insults, or cultural references specific to minority communities are not flagged often enough, so the model fails to detects harmful content in those groups' languages.
   - Relabeling possibility: The minority flags borderline content from their community that the platform's global guidelines ignore.

2. Resume screening
   - Firm and goal: A multi-national HR firm trains a classifier to extract skills from resumes.
   - End-users: Job applicants submitting resumes.
   - Disadvantaged group: Applicants from a disadvantaged minority may lack formal education and degrees compared to the majority, but may have informal training which the classifier ignores.
   - Relabeling possibility: Applicants can reframe their work experience, e.g. framing working at a store as being a salesperson, or managing shifts as managerial experience.

3. Medical treatment prediction
   - Firm and goal: A nationwide insurer builds a treatment-recommendation model to minimize average costs and adverse events.
   - End-users: Patients who report their treatment outcomes (pain levels, recovery time, side effects).
   - Disadvantaged group: Minority groups may experience different side effects or recovery rates than the majority, so the model recommends suboptimal treatments for them.
   - Relabeling possibility: Individual patients record more detailed outcomes rather than underreporting, e.g., consistently marking "still in pain" instead of "fine".

4. Credit scoring
   - Firm and goal: A lender trains a credit-risk model to predict defaults and set loan terms, using historical repayment data.
   - End-users: Borrowers whose repayment or default becomes training labels.
   - Disadvantaged group: Disadvantaged groups may not have credit cards or have never taken loans, and only deal with cash but still pay their bills. These actions are "credit-invisible".
   - Relabeling possibility: A borrower can report their payed bills, such as rent or utilities, as repaid loans. These become additional positive repayment labels.

5. Recommender systems
   - Firm and goal: A streaming platform trains recommender system to maximize engagement, heavily weighted toward mainstream content Baumann & Mendler-Dünner (2024).
   - End-users: Users who like, skip, or re-listen to songs.
   - Disadvantaged group: Niche genres or local musicians get suppressed, as engagement data mostly comes from the majority's preferences.
   - Relabeling possibility: Users can promote underrepresented content by repeatedly listening, liking, or playlisting it.

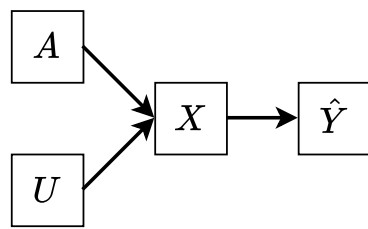

Figure 9: Assumed causal model for data generation and prediction. The group membership $A$ and the other latent variables $U$ are the causal parents of the observable features $X$. The classifier outputs a predicted label $\hat{Y}$ that depends on the features $X$.

## B PRELIMINARIES

### B.1 STATISTICAL PARITY AND EQUALIZED ODDS

Among the various ways fairness can be defined in machine learning, group fairness is one of the most studied. Group fairness requires that a model's predictions should not systematically differ between protected groups. One standard measure of this is statistical parity (SP), which captures the difference in the probability of a positive prediction across groups. Formally, it is defined as

$$\text{SP}(h) = |P[h(x) = 1|a = 1] - P[h(x) = 1|a = 0]|, \tag{12}$$

where a smaller SP value indicates fairer treatment across groups. However, SP does not account for the ground-truth labels $y$, and thus optimizing for SP can degrade the overall accuracy. For example, a classifier that always predicts $\hat{y} = 1$ will have perfect SP but a high prediction error. Alternatively, a stricter notion called equalized odds (EqOd) Hardt et al. (2016) requires that both the true positive rate and false positive rate be equal across groups. Here the EqOd difference is defined as

$$\text{EqOd}(h) = \frac{1}{2} \sum_{z=0,1} |P[h(x) = 1|a = 1, y = z] - P[h(x) = 1|a = 0, y = z]|. \tag{13}$$

### B.2 SUBOPTIMAL BAYES CLASSIFIER

**Definition 2** ($\epsilon$-suboptimal classifier). *A classifier $f : \mathcal{X} \to \mathcal{Y}$ is $\epsilon$-suboptimal on a set $\mathcal{X}' \subseteq \mathcal{X}$ under the distribution $\mathbb{P}$ if there exists a $\mathbb{P}'$ with $\text{TV}\left(\mathbb{P}_{Y|X=x}, \mathbb{P}'_{Y|X=x}\right) \le \epsilon$ such that for all $x \in \mathcal{X}'$*

$$f(x) = \underset{y \in \mathcal{Y}}{\arg\max} \, \mathbb{P}'(y|x).$$

$\text{TV}(\cdot, \cdot)$ is the total variation distance between two distributions. The definition is discussed more in Hardt et al. (2023).

## C THEORETICAL RESULTS AND PROOFS

### C.1 COUNTERFACTUAL FAIRNESS AS SUCCESS

**Proposition 1.** *A Bayes classifier trained on $\mathbb{P}_\alpha$ is minority-focused counterfactually fair if and only if the success of a minority collective is $S = 1$.*

*Proof.* For this proof, we assume the data is generated according to the causal model presented in Figure 9, where the features $X$ are conditioned on the group membership $A$ and other latent causal parent $U$. The features $X$ are then used by a classifier to compute a predicted label $h(x)\hat{Y}$. In our

case, the predicted label is the output of an optimal Bayes classier that predicts the most probable label as $h(x) = \arg\max_y P(y|x)$.

The data distribution is a mixture distribution between the majority distribution $\mathbb{P}_{A=0}$ and the minority distribution $\mathbb{P}_{A=1}$, which is defined as

$$\mathbb{P}_0 = (1-\beta)\,\mathbb{P}_{A=0} + \beta\mathbb{P}_{A=1}, \tag{14}$$

where $\beta$ is the proportion of the minority in the data.

The collective is employing the signal erasure strategy from Equation (4), where the erased signal is the counterfactual of $x$ if they were a member of the majority group $A=0$, or formally as

$$g(x) = x_{A\leftarrow 0} \sim \mathbb{P}(X_{A\leftarrow 0}). \tag{15}$$

The training distribution is a mixture distribution of the data distribution $\mathbb{P}_0$ and the collective distribution $\mathbb{P}^*$, which is defined as

$$\mathbb{P}_\alpha = \alpha\mathbb{P}^* + (1-\alpha)\mathbb{P}_0. \tag{16}$$

We now write the success of the collective (Equation (3)) in terms of the Bayes classifier as

$$
\begin{aligned}
S &= \mathbb{P}_0\left[h(x) = h(g(x))\right] \\
&= \mathbb{P}_0\left[\arg\max_y \mathbb{P}_\alpha(y|x) = \arg\max_y \mathbb{P}_\alpha(y|g(x))\right].
\end{aligned}
\tag{17}
$$

To compute this probability, we split it into two cases, conditioning on the group membership $A$.

When conditioning the success on the majority group $A=0$, then $g(x) = x$ as the intervention on $A$, which converts to the majority, does not change the value of $A$, which is already the majority. This trivially leads to

$$
\begin{aligned}
S_{A=0} &= \mathbb{P}_0\left[\arg\max_y \mathbb{P}_\alpha(y|x, A=0) = \arg\max_y \mathbb{P}_\alpha(y|g(x), A=0)\right] \\
&= \mathbb{P}_0\left[\arg\max_y \mathbb{P}_\alpha(y|x, A=0) = \arg\max_y \mathbb{P}_\alpha(y|x, A=0)\right] \\
&= 1.
\end{aligned}
\tag{18}
$$

For conditioning the success on the minority, recall that the data is generated according to the causal model in Figure 9 which means that intervention on the group membership $A$ can be passed down to the features $X$ as

$$\mathbb{P}(h(x_{A\leftarrow 0}) = y|X, A=1) = \mathbb{P}(h(x) = y|X_{A\leftarrow 0}, A=1) = \mathbb{P}(h(x) = y|g(X), A=1). \tag{19}$$

This can be used to write the success conditioned on the minority as

$$
\begin{aligned}
S_{A=1} &= \mathbb{P}_0\left[\arg\max_y \mathbb{P}_\alpha(y|x, A=1) = \arg\max_y \mathbb{P}_\alpha(y|g(x), A=1)\right] \\
&= \mathbb{P}_0\left[\arg\max_y \mathbb{P}_\alpha(h(x_{A\leftarrow 1}) = y|X, A=1) = \arg\max_y \mathbb{P}_\alpha(h(x_{A\leftarrow 0}) = y|X, A=1)\right].
\end{aligned}
\tag{20}
$$

The first term is rewritten to use the intervention notation even though the intervened variable is unchanged.

As the proportion of the minority is known to be $\beta$, the success can be written by combining Equations (18) and (20) using the law of total probability as

$$
\begin{aligned}
S &= 1 - \beta + \beta\mathbb{P}_0\left[\arg\max_y \mathbb{P}_\alpha(h(x_{A\leftarrow 1}) = y|X, A=1) = \arg\max_y \mathbb{P}_\alpha(h(x_{A\leftarrow 0}) = y|X, A=1)\right] \\
&= 1 - \beta\left(1 - \mathbb{P}_0\left[\arg\max_y \mathbb{P}_\alpha(h(x_{A\leftarrow 1}) = y|X, A=1) = \arg\max_y \mathbb{P}_\alpha(h(x_{A\leftarrow 0}) = y|X, A=1)\right]\right).
\end{aligned}
\tag{21}
$$

This equality can be examined under two scenarios: when the success is perfect $S = 1$ and when the classifier is minority-focused counterfactually fair.

**When the success is** $S = 1$   If the success of the collective is $S = 1$, then Equation (21) leads to

$$\mathbb{P}_0 \left[ \arg \max_y \mathbb{P}_\alpha \left( h \left( x_{A \leftarrow 1} \right) = y | X, A = 1 \right) = \arg \max_y \mathbb{P}_\alpha \left( h \left( x_{A \leftarrow 0} \right) = y | X, A = 1 \right) \right] = 1. \quad (22)$$

This means that it is certain that

$$\arg \max_y \mathbb{P}_\alpha \left( h \left( x_{A \leftarrow 1} \right) = y | X, A = 1 \right) = \arg \max_y \mathbb{P}_\alpha \left( h \left( x_{A \leftarrow 0} \right) = y | X, A = 1 \right), \quad (23)$$

Since the label is binary, then it follows that the same applies to using $\arg \min$. Therefore, for all $y \in \{0, 1\}$ we have

$$\mathbb{P}_\alpha \left( h \left( x_{A \leftarrow 1} \right) = y | X, A = 1 \right) = \mathbb{P}_\alpha \left( h \left( x_{A \leftarrow 0} \right) = y | X, A = 1 \right), \quad (24)$$

which is the definition of a minority-focused counterfactually fair classifier (Definition 1).

**When the classifier is one-sided counterfactually fair**   If the classifier is one-sided counterfactually fair (Definition 1), then by definition

$$\mathbb{P}_0 \left[ \arg \max_y \mathbb{P}_\alpha \left( h \left( x_{A \leftarrow 1} \right) = y | X, A = 1 \right) = \arg \max_y \mathbb{P}_\alpha \left( h \left( x_{A \leftarrow 0} \right) = y | X, A = 1 \right) \right] = 1 \quad (25)$$

and plugging that in Equation (21) results in $S = 1$. $\qquad \square$

### C.2   Impossibility of fairness under ERM

The following proposition follows the structure of Theorem 6 in Chaudhuri et al. (2023). For a vector $x \in \mathbb{R}^d$, let $D(x)$ denote a distribution on $\mathbb{R}^d$ with mean $x$. Let $p$ and $m$ be the number of majority and minority sample, respectively with $p \gg m$.

**Assumption 1** (Concentration Condition, Assumption 2 from Chaudhuri et al. (2023)). *Let* $x_1, \ldots, x_n \overset{i.i.d.}{\sim} D(0)$ *in* $\mathbb{R}^d$. *There exist maps* $X_{\max}, c, C : \mathbb{Z}_+ \times [0, 1] \times \mathbb{Z}_+ \to \mathbb{R}$ *such that for all* $n \geq n_0$, *all* $\delta \in (0, 1)$, *and all unit vectors* $v \in \mathbb{R}^d$, *with probability at least* $1 - \delta$

$$\max_{i \in \{1 \cdots, n\}} \left\{ v^\top x_i \right\} \in \left[ X_{\max}(n, \delta, d) - c(n, \delta, d), X_{\max}(n, \delta, d) + C(n, d, \delta) \right]$$

*and* $\lim_{n \to \infty} C(n, \delta, d) = 0$, $\lim_{n \to \infty} c(n, \delta, d) = 0$.

**Data Model**   Labels $y \in \{-1, 1\}$ and protected attribute $a \in \{-1, 1\}$ define four groups whose class-conditional distributions share the same shape $D(\cdot)$ but have different means:

$$x \mid (y, a) \sim D\big( y\mu + ya\psi \big),$$

where $\mu, \psi \in \mathbb{R}^2$ and $\mu \perp \psi$, with $\hat{\mu} = \mu / \|\mu\|$ and $\hat{\psi} = \psi / \|\psi\|$. For concreteness, take $\mu = \|\mu\| (0, 1)^\top$ and $\psi = \|\psi\| (1, 0)^\top$. Without loss of generality, let the majority attribute be $a_M = +1$ (the minority is $a_m = -1$). Thus the two majority means lie on the positive diagonal $\pm(\mu + \psi)$ and the two minority means on the negative diagonal $\pm(\mu - \psi)$.

Let $B, A$ be two sets of points that are sampled from $D(0)$. We will always associate $B$ with negatively labelled points and $A$ with positive, as will be clear below. Following Chaudhuri et al. (2023), define the sets

$$A_\mu = \{x + \mu : x \in A\}, \qquad -B_\mu = \{x + \mu : x \in -B\}.$$

We split by attribute and (for the minority) allow arbitrary relabeling before training. Write $A_\mu^M$, $B_\mu^M$ for the majority parts and $A_\mu^m, B_\mu^m$ for the minority subsets used with positive/negative labels in training after centering. Incorporating the attribute shifts, set

$$A_{\mu,\psi}^M = -\psi + A_\mu^M, \quad B_{\mu,\psi}^M = +\psi + B_\mu^M, \qquad A_{\mu,\psi}^m = +\psi + A_\mu^m, \quad B_{\mu,\psi}^m = -\psi + B_\mu^m,$$

and similarly for the relabeled minority pieces $A_{\mu,\psi}^{m,\pm}$ and $B_{\mu,\psi}^{m,\pm}$ (these are subsets of $A_{\mu,\psi}^m$ and $B_{\mu,\psi}^m$, respectively).

If the minority were absent, the ERM SVM converges to the spurious direction

$$w_{\text{spu}}^{\text{maj}} \propto \mu + \psi.$$

However, we assume that the minority is performing some relabeling. As a result, we denote the set $A_\mu^{m,+}$ as the samples relabeled with $y = 1$ and the set $A_\mu^{m,-}$ as the samples keeping the original label $y = 0$. Similarly we denote the set $B_\mu^{m,+}$ as the positive minority keeping their labels and $B_\mu^{m,-}$ as the positive minority who flip to $y = 0$.

**Proposition 5.** *Suppose $D(0)$ satisfies Assumption 1 and*

$$X_{max}(p, \delta, 2) - X_{max}(m, \delta, 2) \geq 2\|\psi\| + c(p, \delta, 2) + C(m, \delta, 2). \tag{26}$$

*Then, for any (possibly adversarial) relabeling of minority training examples, if $p \to \infty$, with probability at least $1 - 4\delta$, the SVM ERM solution converges to the same spurious solution $w_{spu}^* \propto \mu + \psi$. Under a centrally symmetric $D(0)$ and when $\|\mu\| = \|\psi\|$, this limit satisfies EqOd $\left(w_{spu}^*\right) \to 0.5$.*

*Proof.* As Chaudhuri et al. (2023) shows, the ERM solution can be written as $w^* = \alpha^*\hat\mu + \sigma\beta^*\hat\psi$, where

$$\alpha^* = \arg\min_{\alpha \in [-1,1], \sigma \in \{-1,1\}} \sup_{x \in \{x|y=0\}} \left(\alpha\hat\mu + \sigma\beta\hat\psi\right)^\mathsf{T}(x - \mu) + \sup_{x \in \{x|y=1\}} \left(\alpha\hat\mu + \sigma\beta\hat\psi\right)^\mathsf{T}(x - \mu)$$

and $\beta = \sqrt{1 - \alpha^2}$.

With the shorthand

$$f_1(\alpha) := \sup_{x \in A_{\mu,\psi}^M} (\alpha\hat\mu + \sigma\beta\hat\psi)^\top x, \qquad f_{2,\pm}(\alpha) := \sup_{x \in A_{\mu,\psi}^{m,\pm}} (\alpha\hat\mu + \sigma\beta\hat\psi)^\top x,$$

$$f_3(\alpha) := \sup_{x \in -B_{\mu,\psi}^M} (\alpha\hat\mu + \sigma\beta\hat\psi)^\top x, \qquad f_{4,\pm}(\alpha) := \sup_{x \in -B_{\mu,\psi}^{m,\pm}} (\alpha\hat\mu + \sigma\beta\hat\psi)^\top x,$$

the SVM objective is

$$F(\alpha) = \min_\alpha \Big\{ \max \big(f_1(\alpha) - \alpha\|\mu\| + \sigma\beta\|\psi\|,$$
$$f(\alpha) - \alpha\|\mu\| - \sigma\beta\|\psi\|,$$
$$f_{4,-}(\alpha) - \alpha\|\mu\| - \sigma\beta\|\psi\|\big)$$
$$+ \max \big(f_3(\alpha) - \alpha\|\mu\| + \sigma\beta\|\psi\|,$$
$$f_{2,+}(\alpha) - \alpha\|\mu\| - \sigma\beta\|\psi\|,$$
$$f_{4,+}(\alpha) - \alpha\|\mu\| - \sigma\beta\|\psi\|\big) \Big\}.$$

By Assumption 1, for the majority group of size $p$, there exists $X_p, c_p, C_p$ such that, with probability at least $1 - 4\delta$ and for all $\alpha$,

$$f_1(\alpha), f_3(\alpha) \in \big[X_p - c_p, X_p + C_p\big]. \tag{27}$$

For any minority relabeling, $A_{\mu,\psi}^{m,\pm} \subseteq A_{\mu,\psi}^m$ and $-B_{\mu,\psi}^{m,\pm} \subseteq -B_{\mu,\psi}^m$, so the same assumption gives

$$f_{2,\pm}(\alpha), f_{4,\pm}(\alpha) \leq X_m + C_m, \tag{28}$$

where $X_m := X_{\max}(m, \delta, 2)$ and $C_m := C(m, \delta, 2)$. Using Equation (26), we get

$$X_p - X_m \geq 2\|\psi\| + c_p + C_m. \tag{29}$$

Combining Equations (27) to (29), still uniformly in $\alpha$, we obtain

$$f_1(\alpha) - f_{2,\pm}(\alpha) \geq 2\|\psi\|, \quad f_1(\alpha) - f_{4,\pm}(\alpha) \geq 2\|\psi\|,$$
$$f_3(\alpha) - f_{2,\pm}(\alpha) \geq 2\|\psi\|, \quad f_3(\alpha) - f_{4,\pm}(\alpha) \geq 2\|\psi\|. \tag{30}$$

**Case 1**: $\sigma = 1$. Consider the *first* inner maximum inside $F(\alpha)$. Compare the majority entry (associated with $f_1(\alpha)$) to the minority entries $(f_{2,-}(\alpha), f_{4,-}(\alpha))$:

$$
\begin{aligned}
[f_1(\alpha) - \alpha \|\mu\| + \beta \|\psi\|] - [f_{2,-}(\alpha) - \alpha \|\mu\| - \beta \|\psi\|] &= (f_1(\alpha) - f_{2,-}(\alpha)) + 2\beta \|\psi\| \\
&\geq 2 \|\psi\| + 2\beta \|\psi\| \\
&\geq 0
\end{aligned}
$$

and similarly against $f_{4,-}$. Hence the first maximum equals $f_1 - \alpha\|\mu\| + \beta\|\psi\|$. For the *second* inner maximum, the same comparison yields the majority term $f_3 - \alpha\|\mu\| + \beta\|\psi\|$. Summing then for $\sigma = 1$ we have,

$$
F_+(\alpha) = f_1(\alpha) + f_3(\alpha) - 2\alpha\|\mu\| + 2\beta\|\psi\|.
$$

**Case 2:** $\sigma = -1$. For the first inner max in $F(\alpha)$,

$$
\begin{aligned}
[f_1(\alpha) - \alpha \|\mu\| - \beta \|\psi\|] - [f_{2,-}(\alpha) - \alpha \|\mu\| + \beta \|\psi\|] &= (f_1(\alpha) - f_{2,-}(\alpha)) - 2\beta \|\psi\| \\
&\geq 2 \|\psi\| - 2\beta \|\psi\| \\
&\geq 0
\end{aligned}
$$

and likewise against $f_{4,-}$. Thus the first maximum equals $f_1 - \alpha\|\mu\| - \beta\|\psi\|$. The second inner max is analogous and equals $f_3 - \alpha\|\mu\| - \beta\|\psi\|$. Therefore,

$$
F_-(\alpha) = f_1(\alpha) + f_3(\alpha) - 2\alpha\|\mu\| - 2\beta\|\psi\|.
$$

For every $\alpha$, $F_+(\alpha) = (f_1 + f_3) - 2\alpha\|\mu\| + 2\beta\|\psi\|$ and $F_-(\alpha) = (f_1 + f_3) - 2\alpha\|\mu\| - 2\beta\|\psi\|$, so $F_+(\alpha) \geq F_-(\alpha)$. Hence the optimal sign is $\sigma = -1$ and the objective reduces to

$$
F(\alpha) = (f_1(\alpha) + f_3(\alpha)) - 2\alpha\|\mu\| - 2\beta\|\psi\|.
$$

Maximizing $\alpha\|\mu\| + \beta\|\psi\|$ is equivalent to minimizing $F(\alpha)$ up to the bounded change (due to Assumption 1) of $f_1(\alpha) + f_3(\alpha)$. Next, we use the following lemma.

**Lemma 1** (Approximate Maximization Lemma - I, Lemma 14 from Chaudhuri et al. (2023)). *Let $F(\alpha) = f(\alpha) + g(\alpha)$ where $g(\alpha) = \alpha u + \sqrt{1 - \alpha^2} v$, $u, v > 0$, and $f(\alpha) \in [-L, U]$. Let $\alpha_F \in \text{argmax}_\alpha F(\alpha)$, and let $\alpha_g = \frac{u}{\sqrt{u^2+v^2}} \in \text{argmax}_\alpha g(\alpha)$.*

*Then, the angle between $(\alpha_F, \sqrt{1-\alpha_F^2})$ and $(\alpha_g, \sqrt{1-\alpha_g^2})$ is at most $\cos^{-1}\left(1 - \frac{L+U}{\sqrt{u^2+v^2}}\right)$, and $\max_\alpha F(\alpha) \geq \sqrt{u^2 + v^2} - L$.*

Applying Lemma 1 with $u = \|\mu\|$ and $v = \|\psi\|$ shows that $(\alpha, \beta)$ approaches

$$
(\alpha_g, \beta_g) = \left( \frac{\|\mu\|}{\sqrt{\|\mu\|^2+\|\psi\|^2}}, \ \frac{\|\psi\|}{\sqrt{\|\mu\|^2+\|\psi\|^2}} \right)
$$

as $p \to \infty$. Thus

$$
w^* \longrightarrow w^*_{\text{spu}} = \alpha_g \hat{\mu} + \beta_g \hat{\psi},
$$

independently of how the minority samples were relabeled in training.

Under a centrally symmetric $D(0)$ and if $\|\mu\| = \|\psi\|$, the majority group ($a = +1$) separates perfectly in the limit, while the minority group ($a = -1$) has symmetric measure about the threshold, giving $\text{TPR}_{a=+1} \to 1$, $\text{FPR}_{a=+1} \to 0$, and $\text{TPR}_{a=-1} = \text{FPR}_{a=-1} \to \frac{1}{2}$. Hence $\text{EqOd}(w^*_{\text{spu}}) \to 0.5$.

$\square$

This result can also be extended to $\mathbb{R}^d$ using techniques similar to those in Chaudhuri et al. (2023). This result also encompasses the 4-Gaussian mixture model $\mathbb{P}_{\text{4GMM}}$ used in Section 5 as a special case, leading to the following.

**Proposition 2** (Informal). *Consider the dataset $\mathbb{P}_{\text{4GMM}}$, where every minority point participates in the ACA by flipping all $y = 0$ labels to $y = 1$. Then, under sufficiently separable clusters, with high probability, the EqOd of the ERM classifier minimizing the logistic loss will asymptotically approach 0.5.*

## C.3 Success Bound With Label Error

The following proof uses Lemma 11 from Hardt et al. (2023).

**Lemma 2** (Lemma 11 from Hardt et al. (2023)). *Suppose that $P, P'$ are two distributions such that $\mathrm{TV}(P, P') \leq \epsilon$. Take any two events $E_1, E_2$ measurable under $P, P'$. If $P(E_1) > P(E_2) + \frac{\epsilon}{1-\epsilon}$, then $P'(E_1) > P'(E_2)$.*

**Proposition 3.** *With algorithm $\mathcal{A}(x)$ with label error $\rho$, the success of the collective is bounded by*

$$S(\alpha) \geq 1 - \frac{2(1-\alpha)}{(1-2\rho)\alpha}\tau - \frac{\epsilon}{(1-\epsilon)(1-2\rho)\alpha}. \tag{11}$$

*Proof.* This proof follows closely the proof of Theorem 5 by Hardt et al. (2023). We start under the assumption of an optimal Bayes classifier, setting $\epsilon = 0$.

When the new label $y'$ is wrong with probability $\rho$, then we can think of the collective as being union of two sub-collectives: one with the correct label and one with the incorrect label. In the binary case this can be formulated with correct subcollective $P^+$ as having label $y' = \arg\max_y P_0(y|g(x))$ and the incorrect subcollective $P^-$ as with label $y' = \arg\min_y P_0(y|g(x))$. Then we can write the train distribution as

$$
\begin{aligned}
P_\alpha &= \alpha\left(\rho P^- + (1-\rho)P^+\right) + (1-\alpha)P_0 \\
&= \alpha\rho P^- + (1-\rho)\alpha P^+ + (1-\alpha)P_0.
\end{aligned} \tag{31}
$$

Denote $y^*(x) = \arg\max_y P_0(y|g(x))$, then the probability to get prediction $y^*$ is

$$
\begin{aligned}
P_\alpha(y^*|x) &= \alpha\rho P^-(y^*|x) + (1-\rho)\alpha P^+(y^*|x) + (1-\alpha)P_0(y^*|x) \\
&= (1-\rho)\alpha + (1-\alpha)P_0(y^*|x),
\end{aligned} \tag{32}
$$

and the probability to get the prediction $y \neq y^*$ is

$$
\begin{aligned}
P_\alpha(y|x) &= \alpha\rho P^-(y|x) + (1-\rho)\alpha P^+(y|x) + (1-\alpha)P_0(y|x) \\
&= \alpha\rho + (1-\alpha)P_0(y|x),
\end{aligned} \tag{33}
$$

where $P^+(y^*|x) = 1$, $P^-(y^*|x) = 0$, $P^+(y^*|x) = 0$, $P^-(y^*|x) = 1$ by definition.

A Bayes classifier $h$ returns the most probable label $h(x) = \arg\max_y P(y|x)$. Therefore, a Bayes classifier will output $y^*$ if the probability is greater, which can be written as the condition

$$
\begin{aligned}
P_\alpha(y^*|x) &> P_\alpha(y|x) \\
(1-\rho)\alpha + (1-\alpha)P_0(y^*|x) &> \alpha\rho + (1-\alpha)P_0(y|x) \\
(1-2\rho)\alpha &> (1-\alpha)(P_0(y|x) - P_0(y^*|x)).
\end{aligned} \tag{34}
$$

Let $\tau(x) = \max_y [P_0(y|x) - P_0(y|g(x))]$, then

$$
\begin{aligned}
P_0(y|x) - P_0(y^*|x) &\leq P_0(y|x) - P_0(y|g(x)) + P_0(y^*|g(x)) - P_0(y^*|x) \\
&\leq 2\tau(x).
\end{aligned} \tag{35}
$$

With that, the condition in Equation (34) can be written as

$$(1-2\rho)\alpha > 2(1-\alpha)\tau(x). \tag{36}$$

With that, the success can be bounded as

$$
\begin{aligned}
S &= P_0[f(x) = f(g(x))] \\
&= P_0[f(x) = y^*(x)] \\
&\geq P_0[(1-2\rho)\alpha > 2(1-\alpha)\tau(x)] \\
&= P_0\left[1 - \frac{2(1-\alpha)}{(1-2\rho)\alpha}\tau(x) > 0\right] \\
&= \mathbb{E}_{x\sim P_0}\left[\mathbf{1}\left\{1 - \frac{2(1-\alpha)}{(1-2\rho)\alpha}\tau(x) > 0\right\}\right] \\
&\geq \mathbb{E}_{x\sim P_0}\left[1 - \frac{2(1-\alpha)}{(1-2\rho)\alpha}\tau(x)\right] \\
&= 1 - \frac{2(1-\alpha)}{(1-2\rho)\alpha}\tau
\end{aligned} \tag{37}
$$

**With sub-optimality** $\epsilon > 0$    A result of Lemma 2 is to write the condition in Equation (36) as

$$(1 - 2\rho)\,\alpha > 2\,(1 - \alpha)\,\tau\,(x) + \frac{\epsilon}{1 - \epsilon}, \tag{38}$$

which by following the same steps as with $\epsilon = 0$ results in the final bound

$$S\,(\alpha) \geq 1 - \frac{2\,(1 - \alpha)}{(1 - 2\rho)\,\alpha}\tau - \frac{\epsilon}{(1 - \epsilon)\,(1 - 2\rho)\,\alpha}. \tag{39}$$

$\square$

### C.4    LABEL ERROR WITH BETTER REPRESENTATION

For the following we assume a similar setting as in Appendix C.2, visualised as a 2D distribution in Figure 6. We are given the majority data, and tasked with labeling the minority data. Assume all labels are distributed equally $\mathbb{P}[Y = 1] = \mathbb{P}[Y = -1] = \frac{1}{2}$. The minority features $X_{\min}$ are distributed as $X_{\min} \sim \mathcal{N}(y\mu_{\min}, \Sigma_{\min})$ with $X_{\min} \in \mathbb{R}^d$. The label $\hat{y}_{1\mathrm{NN}}^{(n)}$ is predicted according to a 1NN classifier from $n$ majority samples $\mathcal{D}_n = (x_i, y_i)_{i=0}^n$. Majority samples with $y = +1$ are distributed as $X_+ \sim \mathcal{N}(\mu, \Sigma)$, and with $y = +1$ are distributed as $X_- \sim \mathcal{N}(-\mu, \Sigma)$.

**Theorem 1.** *Assume that $\mu_{min}^\top \Sigma^{-1}\mu > 0$. Further, consider the setting with $\Sigma_{min} = I$, and the minority (i.e. test) distribution introduced above with $\mathbb{P}[Y = 1] = \mathbb{P}[Y = -1] = 0.5$ and $X_{min} \sim \mathcal{N}(y\mu_{min}, \Sigma_{min})$.*

*Then, there exists a projection $P \in \mathbb{R}^{d \times d}$ such that asymptotically for $n \to \infty$, $err_{1NN}^{rep} < err_{1NN}^{raw}$.*

*Proof.* Consider the projection on the hyperplane perpendicular to $w$, where $w = \frac{\mu - \mu_{\min}}{2}$. The projection matrix associated with this transformation is $P = I - \frac{ww^\top}{w^\top w}$.

Let us denote the symbols after the projection as $\bar{\mu} := P\mu$, $\bar{\mu}_{\min} := P\mu_{\min}$, $\bar{v} := (P\Sigma P^T)^+ \bar{\mu}$ and $\bar{\Sigma}_{\min} := P\Sigma_{\min}P^T$. Here we denoted using $A^+$ the pseudoinverse of the matrix $A$. Note that since $P$ is an orthogonal projection matrix, it holds that $PP = P$ and $P^T = P$.

We apply Lemma 3 to obtain closed forms for the asymptotic error of 1NN applied to the initial representation and to the features after the projection $P$. Namely, using the notation $v := \Sigma^{-1}\mu$ we have:

$$\mathrm{err}_{1\mathrm{NN}} = \frac{1}{2}\mathbb{P}_{X_{\min}|y=1}[\hat{y}_{1\mathrm{NN}} = -1] + \frac{1}{2}\mathbb{P}_{X_{\min}|y=-1}[\hat{y}_{1\mathrm{NN}} = 1] \tag{40}$$

$$= \frac{1}{2}\left(1 - \Phi\left(\frac{v^\top \mu_{\min}}{\sqrt{v^\top \Sigma_{\min}v}}\right)\right) + \frac{1}{2}\Phi\left(\frac{-v^\top \mu_{\min}}{\sqrt{v^\top \Sigma_{\min}v}}\right) \tag{41}$$

$$= 1 - \Phi\left(\frac{v^\top \mu_{\min}}{\sqrt{v^\top \Sigma_{\min}v}}\right) \tag{42}$$

$$= 1 - \Phi(SNR), \tag{43}$$

where we used the fact that $\Phi(-z) = 1 - \Phi(z)$ and we denote $SNR := \frac{v^\top \mu_{\min}}{\sqrt{v^\top \Sigma_{\min}v}}$.

Similarly, let us denote the SNR corresponding to 1NN applied on the projected representation as follows: $SNR_{\mathrm{proj}} := \frac{\bar{v}^\top \bar{\mu}_{\min}}{\sqrt{\bar{v}^\top \bar{\Sigma}_{\min}\bar{v}}}$.

To show that $\mathrm{err}_{1\mathrm{NN}} > \mathrm{err}_{1\mathrm{NN}}^{\mathrm{rep}}$ it suffices to prove that $SNR < SNR_{\mathrm{proj}}$.

We begin by rewriting the numerator of $SNR_{\mathrm{proj}}$. Since $\mu \in Im(P)$ and because on $Im(P)$ the operators $\Sigma^{-1}$ and $(P\Sigma P^\top)^+$ represent the same transformation, it follows that:

$$\bar{v} = (P\Sigma P^\top)^+ \bar{\mu} = \Sigma^{-1}\bar{\mu}.$$

Moving on the the denominator of $SNR_{\text{proj}}$, we have that:

$$
\begin{aligned}
\bar{v}^\top \bar{\Sigma}_{\min} \bar{v} &= \bar{v}^\top (P \Sigma_{\min} P^\top)^+ \bar{v} \\
&= \bar{v}^\top (PP^\top)^+ \bar{v} \\
&= \bar{v}^\top P^+ \bar{v} \\
&= \bar{v}^\top P \bar{v} \\
&= \bar{v}^\top \bar{v} \\
&= \|\bar{v}\|^2.
\end{aligned}
$$

In the second line we used the fact that $\Sigma_{\min} = I$, in the third line we use the identity $P^2 = P$ due to $P$ being a projection matrix, in the forth line we use $P^+ = P$ since $P$ is an orthogonal projection (i.e. $P$ is symmetric) and in the fifth line we use the fact that $\bar{v} \in Im(P)$, and hence, $P\bar{v} = \bar{v}$.

Putting everything together, and using the fact that $\Sigma$ (and thus, $\Sigma^{-1}$) is positive definite (i.e. $x^\top \Sigma^{-1} x > 0, \forall x \in \mathbb{R}^d$) we get that:

$$
SNR_{\text{proj}} = \frac{\bar{\mu}^\top \Sigma^{-1} \bar{\mu}}{\|\bar{v}\|^2} > 0 > \frac{\mu^\top \Sigma^{-1} \mu_{\min}}{\|\Sigma^{-1} \mu\|^2} = SNR.
$$

$\square$

**Lemma 3.** *For a unimodal minority distribution $X_{min} \sim \mathcal{N}(\mu_{min}, \Sigma_{min})$ it holds that:*

$$
\lim_{n \to \infty} \mathbb{P}_{X_{min}}[\hat{y}_{1NN}^{(n)} = -1] = 1 - \Phi\left( \frac{v^\top \mu_{min}}{\sqrt{v^\top \Sigma_{min} v}} \right),
$$

*where $v := \mu^\top \Sigma^{-1}$ and $\Phi$ is the CDF of a standard Gaussian.*

*Proof.* Let us denote $\hat{y}_{1NN} := \lim_{n \to \infty} \hat{y}_{1NN}^{(n)}$ and let $p_+$ and $p_-$ be the densities of two class-conditional distribution. Notice that the two class conditional training distributions are supported on the entire domain of $\mathbb{R}^d$. Therefore, in the asymptotic regime, the label $\hat{y}_{1NN}$ at a test point $x$ is given according to the class-conditional distribution that has higher density. Namely, we have:

$$
\hat{y}_{1NN} = \begin{cases} -1 & \text{if } p_+(x) < p_-(x), \\ 1 & \text{otherwise.} \end{cases}
$$

Given $X_{\min} \sim \mathcal{N}(\mu_{\min}, \Sigma_{\min})$, we can then write the probability of predicting $\hat{y}_{1NN} = -1$ as:

$$
\mathbb{P}_{X_{\min}}[\hat{y}_{1NN} = -1] = \mathbb{P}_{X_{\min}}[p_+(x) < p_-(x)].
$$

Using the closed forms for the pdf of a Gaussian, we write the corresponding log-probabilities as follows:

$$
\log p_+(x) = -\frac{1}{2}(x - \mu)^\top \Sigma^{-1}(x - \mu) + \text{const.}
$$

$$
\log p_-(x) = -\frac{1}{2}(x + \mu)^\top \Sigma^{-1}(x + \mu) + \text{const.}
$$

Using the fact that $\log$ is monotonically increasing and $\Sigma$ (and by extension $\Sigma^{-1}$) is a symmetric matrix, we can write after some simple calculations:

$$
\mathbb{P}_{X_{\min}}[\hat{y}_{1NN} = -1] = \mathbb{P}_{X_{\min}}[\mu^\top \Sigma^{-1} x < 0].
$$

Let us denote the random variable $Z := (\mu \Sigma^{-1})X$. Since $Z$ is a linear transformation of Gaussian random variable, it is itself Gaussian and we can write its mean and variance as follows:

$$\mu_Z := v^\top \mu_{\min}, \text{ and } \sigma_Z^2 := v^\top \Sigma_{\min} v, \text{ where } v := \mu^\top \Sigma^{-1}.$$

After this change of variable, we can rewrite the probability of predicting $\hat{y}_{1NN} = -1$ as:

$$\begin{aligned}
\mathbb{P}_{X_{\min}}[\hat{y}_{1NN} = -1] &= \mathbb{P}_Z\left[Z < 0\right] \\
&= \Phi\left(\frac{0 - \mathbb{E}[Z]}{\sqrt{\mathrm{Var}[Z]}}\right) \\
&= \Phi\left(\frac{-(\mu^\top \Sigma^{-1})^\top \mu_{\min}}{\sqrt{(\mu^\top \Sigma^{-1})^\top \Sigma_{\min}(\mu^\top \Sigma^{-1})}}\right) \\
&= 1 - \Phi\left(\frac{(\mu^\top \Sigma^{-1})^\top \mu_{\min}}{\sqrt{(\mu^\top \Sigma^{-1})^\top \Sigma_{\min}(\mu^\top \Sigma^{-1})}}\right).
\end{aligned}$$

$\square$

Note that the error from Theorem 1 is defined the same as $\rho$ (Equation (10). This leads to the following.

**Proposition 4** (Informal). *Let data be drawn from $\mathbb{P}_{4GMM}$, and $\rho_{plain}$ denote the error of a 1-NN classifier that assigns the label of the nearest majority neighbor in the original feature space. Then there exists a fair representation in which a 1-NN classifier achieves error $\rho_{FRL}$ such that, asymptotically with respect to the dataset size, $\rho_{FRL} \le \rho_{plain}$.*

## D    TECHNICAL DETAILS

### D.1    DATASETS

**COMPAS**    The Correctional Offender Management Profiling for Alternative Sanctions (COMPAS) dataset contains the data of criminal defendants in Broward county sheriff's office in Florida with the task of predicting the recidivism risk. The label in this dataset represents whether the person re-offended and the sensitive attribute is the race. We follow the same data cleaning and pre-processing as Alghamdi et al. (2022).

**Adult**    The Adult dataset Becker & Kohavi (1996) contains demographic features of US citizens and is tasked with predicting the income level of an individual. The label represents if the individual has income higher than $50,000 and the sensitive attribute we use is the race. We follow the same data cleaning and pre-processing as Alghamdi et al. (2022).

**HSLS**    The High School Longitudinal Study of 2009 (HSLS) Jeong et al. (2022) contains details of high-school students across the US and the task is to predict the academic success of the students. The label represents the exam score and the sensitive attribute is the race. We follow the same data cleaning and pre-processing as Alghamdi et al. (2022).

**ACS-Income**    Ding et al. (2021) offer different classification tasks derived by US census data. In our work we used the pre-defined task of predicting level of income denoted as *ACSIncome*, where the data is already pre-processed. The label represents if the individual has income higher than $50,000 and the sensitive attribute is the race.

**Waterbirds**    The waterbirds dataset Sagawa et al. (2020) contains images of landbirds and water-birds super-imposed on either land or water backgrounds, with the task of classifying the image as of a landbird or a waterbird. The label represents the type of bird, and the sensitive attribute is whether the background is land or water. To obtain the features, we used the output of the penultimate layer of a pre-trained ResNet-18 network from *PyTorch* [1]. We report the results on those features

---

[1] https://pytorch.org/vision/main/models/generated/torchvision.models.resnet18.html

as Waterbirds-Full. We also performed PCA (using *scikit-learn*) and kept the first 85 principal components which retain about 75% of the variance, and report the results of these components as Waterbirds-PCA.

**CivilComments**   The CivilComments dataset Borkan et al. (2019) is a collection of text comments found on the internet, with the goal of training a classifier to fairly detect toxicity. For this paper, we modified the dataset to keep only the comments that include either *christian* or *muslim* (but not both), with a label 0 meaning toxic and 1 meaning safe. To obtain the features, we used the word embeddings given by Hugging Face's *bert-base-uncased* model[2]. We report the results on those features as CivilComments-Full. We also performed PCA (using *scikit-learn*) and kept the first 100 principal components which retain about 75% of the variance, and report the results of these components as CivilComments-PCA.

## D.2   TRAINING

All classification experiments were trained with *scikit-learn*'s histogram-based gradient boosting classification tree with the default parameters [3]. When there was not a pre-defined test set, we set the train-test split as 80-20 before applying the collective action.

The probabilities for RB-prob were inferred by training *scikit-learn*'s histogram-based gradient boosting classification tree on the majority data with the default parameters, and using its *predict_proba* function. For LFR Zemel et al. (2013) we used the implementation in *Holistic AI*'s open source library [4] with the default parameters. For FARE Jovanović et al. (2023) we used the official implementation [5] with hyperparameters $\gamma = 0.85$, $k = 200$ and $n = 100$. For all distance computation we used the Euclidean norm $\ell^2$-norm as $d(v, u) = \|v - u\|_2 = \sqrt{\sum_i (v_i - u_i)^2}$.

# E   ADDITIONAL RESULTS

## E.1   COMPARISON WITH PRIOR WORK

We compare our method RB-prob with the existing methods KDP Luong et al. (2011) and CND Kamiran & Calders (2009) in Figure 10. Note than CND requires flipping labels for both majority and minority members, and we report the total number of label flips. Figure 10 shows that our method, motivated by the counterfactual labeling, is more efficient in terms of required number of label flips, than the existing works.

## E.2   EXPANDED RESULTS

The following figures include the results of the experiments reported in the main text using all methods on all dataset, both with EqOd (Equation (2)) and SP (Equation (12)) as a measure of unfairness

---

[2]https://huggingface.co/google-bert/bert-base-uncased

[3]scikit-learn.org/stable/modules/generated/sklearn.ensemble.HistGradientBoostingClassifier.html

[4]https://github.com/holistic-ai/holisticai

[5]https://github.com/eth-sri/fare

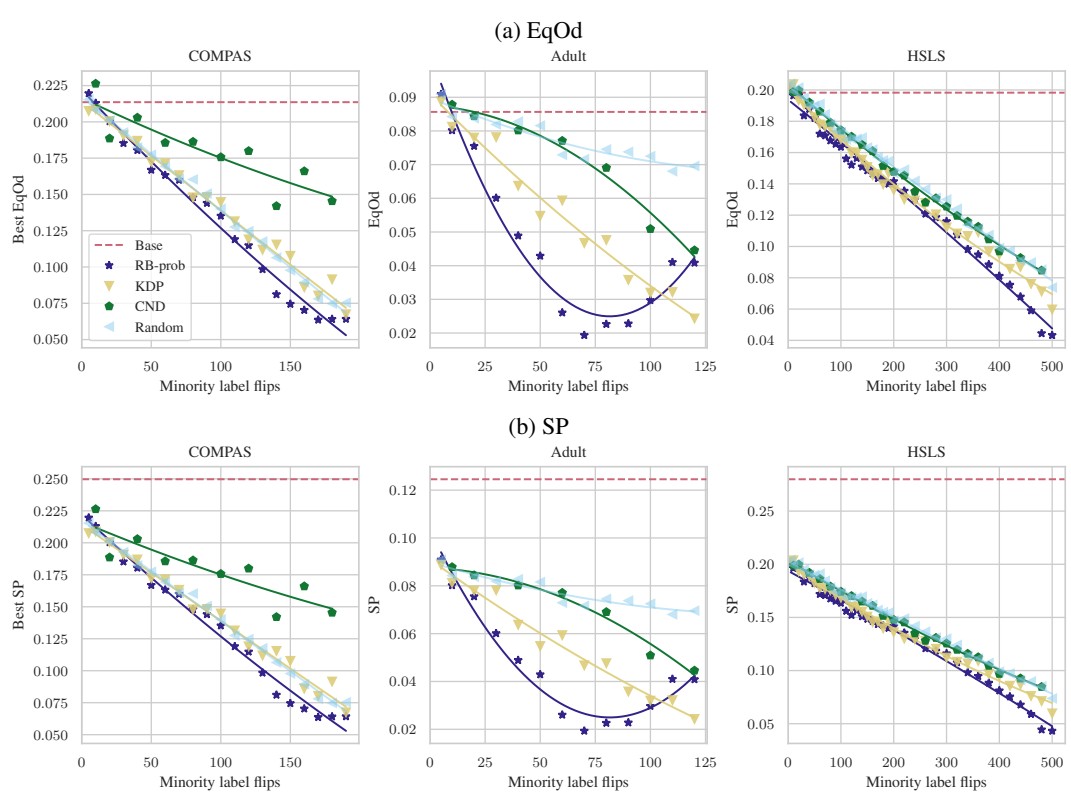

Figure 10: Fairness per number of label flips of the Random baseline, our method RB-prob, and the existing methods KDP (Luong et al., 2011) and CND (Kamiran & Calders, 2009). Our method is more efficient than prior work, requiring less flips to achieve the same level of fairness. Note that in this experiment CND could flip any label, while all other methods were restricted to the labels of 30% of the minority.

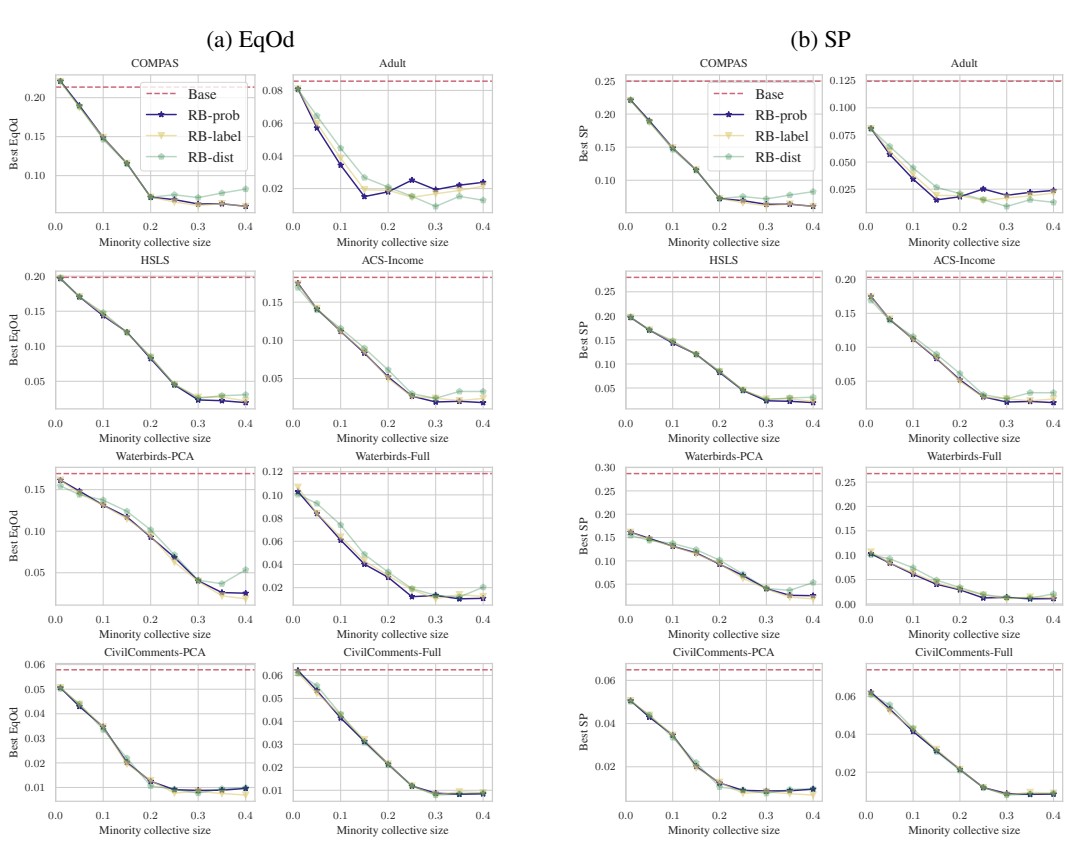

Figure 11: The lowest EqOd violation a collective can achieve greatly improves as the collective size increases, up to a certain point. Each point is a mean of 10 runs, with the standard deviation being smaller than the markers. In all the datasets we experimented on, the lowest EqOd violation converges around $\alpha = 0.3$.

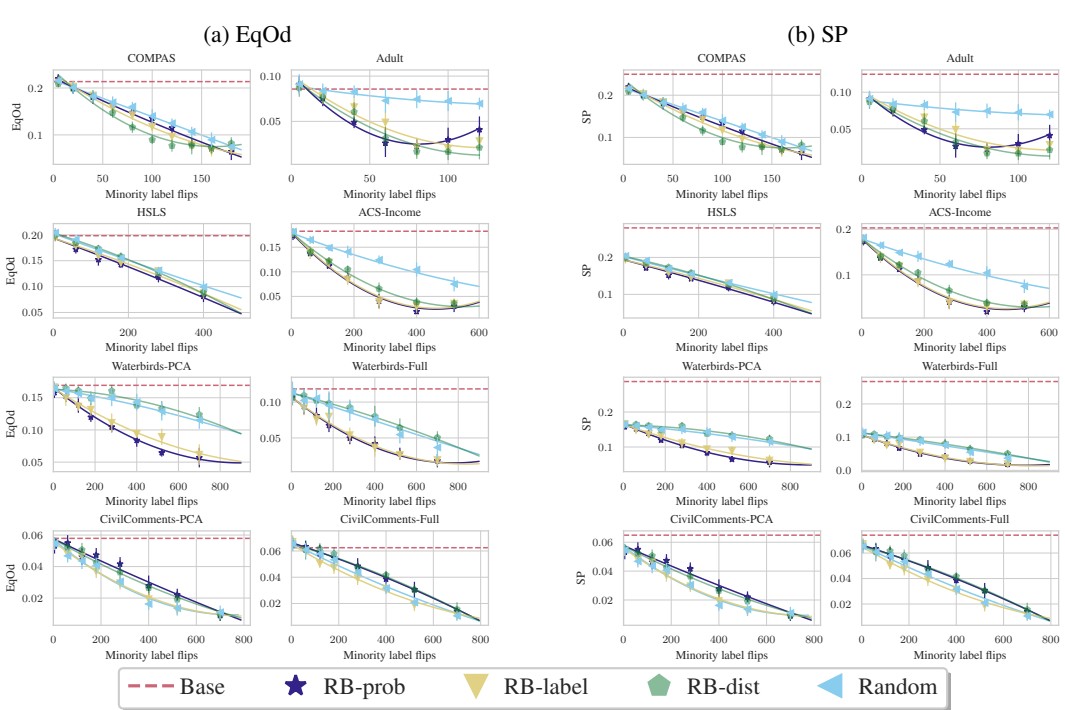

Figure 12: Our proposed methods are consistently more efficient than randomly flipping labels, requiring less label flips to attain the same level of EqOd. Each marker is the mean of 10 random runs with a specific number of label flips. The dashed line shows the mean EqOd for a classifier trained on the dataset without collective action.

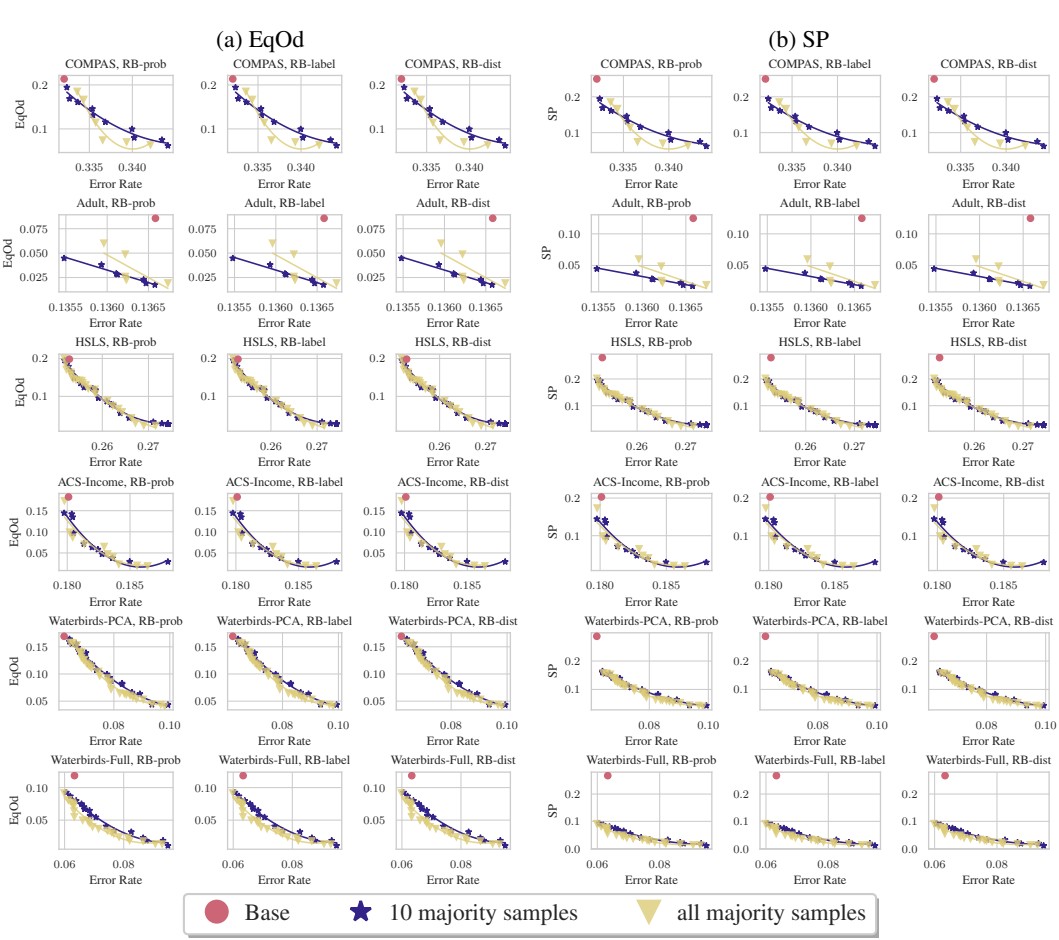

Figure 13: Limiting the knowledge of the collective about the majority does not significantly harm the Pareto front. Each point is the mean of 10 runs and the curves are fitted to guide the eye.

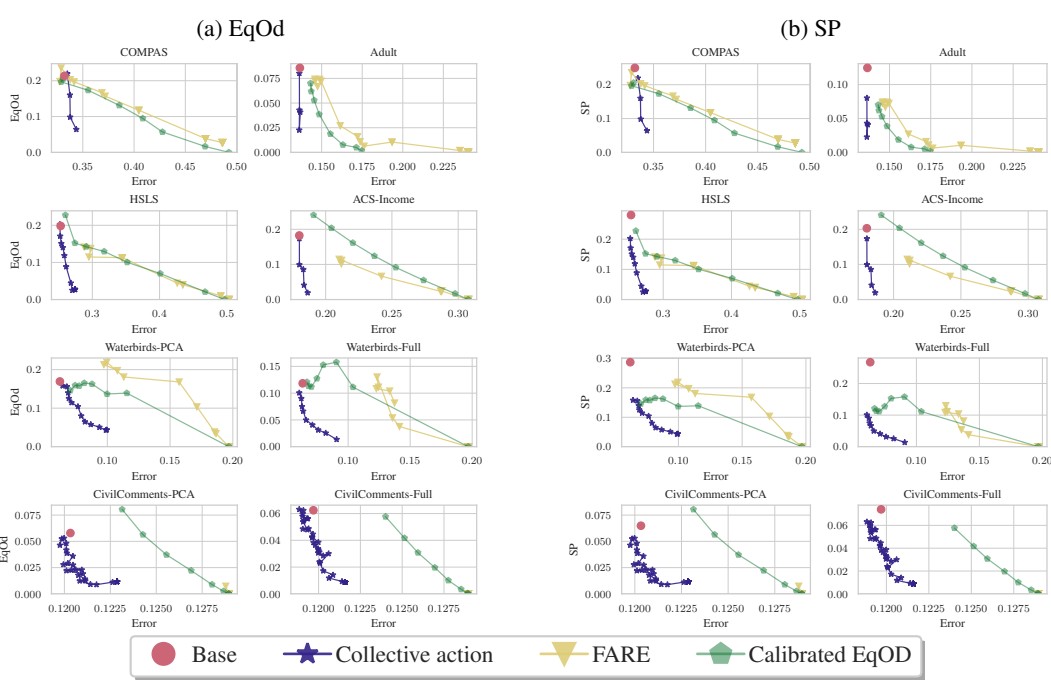

Figure 14: The firm-side pre-processing method FARE Jovanović et al. (2023) and the post-processing method calibrated equalized odds Pleiss et al. (2017) attain 0 EqOd with large error, while RB-prob with $\alpha = 0.3$ (Section 3) has much smaller error and less unfairness than the base classifier, but unable to get 0 EqOd.

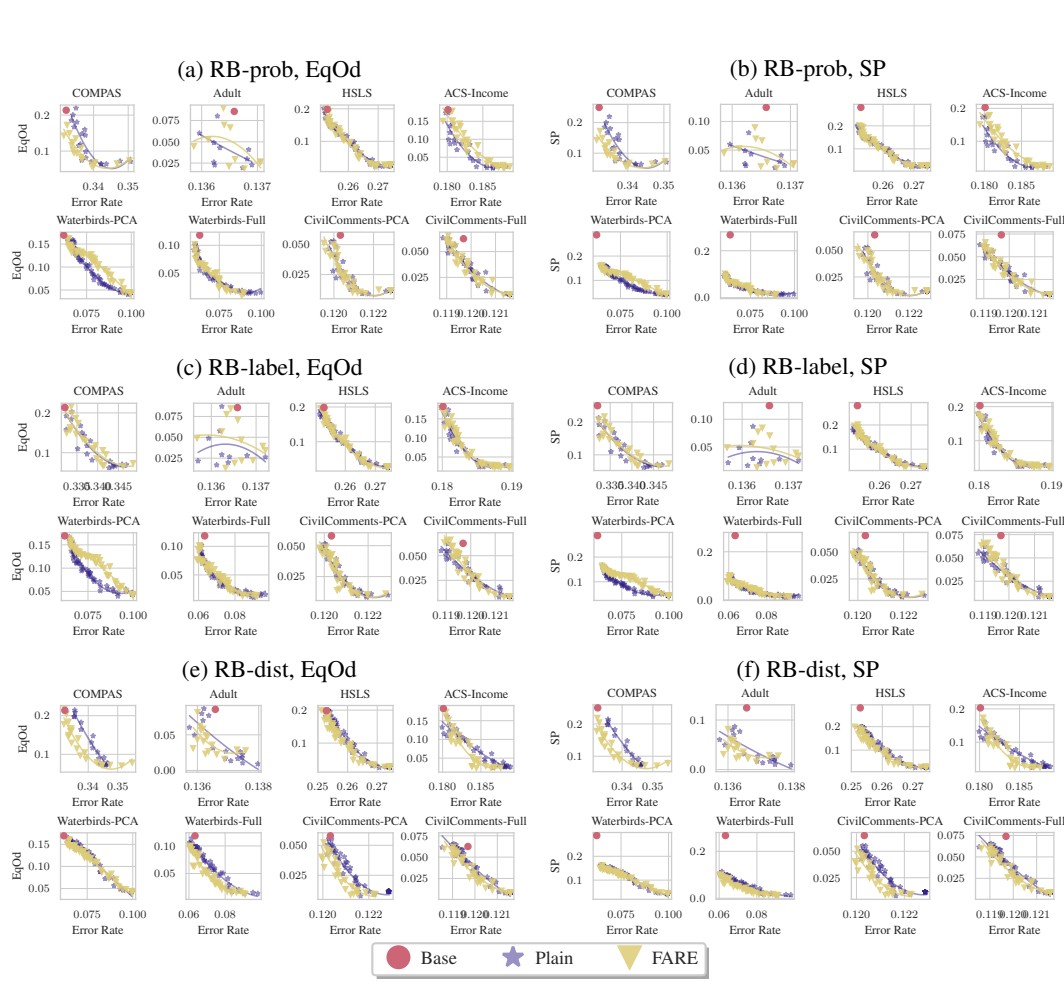

Figure 15: The Pareto fronts for using a fair representation when computing the KNN for RB-dist dominate the Pareto fronts for KNN computed on untransformed features. The blue stars represent the KNN without transforming the data, and the yellow triangles represent the KNN when the data is transformed using FARE (Jovanović et al., 2023). The lines are fitted by a polynomial of degree 2 to guide the eye.

