# OpenReview forum: "Fairness for the People, by the People: Minority Collective Action"
_ICLR.cc/2026/Conference — Submitted to ICLR 2026_

### Official Review · Reviewer_YvuG · 2025-10-14

**Soundness:** 2
**Presentation:** 3
**Contribution:** 2
**Rating:** 4
**Confidence:** 3

**Summary:**

This paper proposes a user-side fairness approach where minority users collectively relabel their data (i.e., Algorithmic Collective Action, ACA) to improve fairness without the decision-maker's intervention. It introduces three "model-agnostic" relabeling methods and shows they significantly reduce unfairness with minimal accuracy loss. The work also analyzes theoretical limits when only minority members act collectively. Experiments show the effectiveness and limits of the method.

**Strengths:**

1. Clarity is great. The paper is generally well-written and easy to understand.
2. The discussions on both how ACA can help with fairness and the limitations of ACA are extensive.
3. The connections to counterfactual fairness make sense to me.
3. The experiments can prove what the authors claim.

**Weaknesses:**

My main concerns lie in the settings:

1. Though I am familiar with the ACA proposed in [Hardt et al., 2023], I am not sure whether it is practical for the minority group collaborating to enhance the fairness of decision-making. In your design, at least all minority group members should share information and then select some members to flip their labels. Could you outline some examples in reality?

2. I am a bit confused about section 3 of approximating the counterfactual label. The minority group members seem to need to have access to lots of information. For RB-prob, they need to know their scores of being admitted; for RB-label, they need to know their neighbors' labels; for RB-dist, it might be more practical to only know the distances.

- Minor weaknesses on related works:

In fact, the setting formulated is very similar to strategic classification (SC) / performative prediction (PP) where the population move their features in response to the deployed models. Thus, it is worthwhile discussing the literature on these topics and the fairness regarding these settings. In my perspective, you are focusing on a setting where the objective of strategic agents is to enhance the fairness, which can be novel. However, I still feel it necessary to review some prominent works on SC/PP. Examples might be [1,2,3], where [1] and [2] are works proposing these frameworks, and [3] is a recent work on fairness in these settings.

[1] Hardt, Moritz, et al. "Strategic classification." Proceedings of the 2016 ACM conference on innovations in theoretical computer science. 2016.

[2] Perdomo, Juan, et al. "Performative prediction." International Conference on Machine Learning. PMLR, 2020.

[3] Jin, Kun, et al. "Addressing polarization and unfairness in performative prediction." arXiv preprint arXiv:2406.16756 (2024).

**Questions:**

See weaknesses 1, 2.

---

> ### Author Response · Authors · 2025-11-21
>
> We thank the reviewer for their thoughtful assessment and helpful suggestions. We are glad that the clarity, experimental support, and discussion of limitations were appreciated.
>
> 1.  **Practicality of ACA**: We would first like to mention that we already discuss several motivating practical examples in Appendix A. Having said that,
>
>
> 	**a)**  We indeed agree with the reviewer that the question of practical collective coordination is an important question in ACA and is not just a problem that needs to be solved for our work but for the entire ACA literature.
>
> 	**b)**  We also consider the coordination to be one of the true costs of ACA and thus, show in Figure 3, that even with a very small collective size ($\alpha=0.3$ means *only 30% of the minority*), the collective needs to coordinate and choose $M$ users among them to flip their labels. In short, our experiments do not assume full participation of the minority. We note in lines 294-295 that all reported results use 30% of the minority. While 30% is still a big part, empirical sociology works show that discrimination motivates collective action of minorities [1,2]. We briefly mention these works in the introduction and conclusion.
>
>
> 2.  **Access to information**: The reviewer’s observation is right that the more a collective knows, the more it can reduce unfairness. However, an important observation of our work (Figures 5, 13), is that if the collective can only access other collective members' information and very few samples from the majority, it still remains substantially effective. We believe an interesting follow-up from our work will be to understand which of the different strategies will be socially more acceptable, however we think that this is beyond the scope of this work. .
>
> 3.  **Differences between algorithmic collective action (ACA), performative prediction (PP) and strategic classification (SC)**: While ACA, PP and SC share similarities they are also different. SC asks how a single user can change their data to maximize their own reward given a trained classifier. However in ACA there is no apriori trained classifier, and the collective works together to affect the population behaviour of a classifier that is about to be learned. A collective is needed in this case because a single person is not statistically strong enough to impact the training algorithm.
>     PP studies how a firm affects the future training data distribution by training a new classifier. This can be caused naturally, by how strategic agents react to it (SC), or by a changed strategy of a collective in case the firm is re-training their model. However, the change in the data distribution in PP is not actively decided by the minority collective but rather passively induced by the firm’s training algorithm. As pointed out by the reviewer, the paper by Jin et al. claims that PP induces unfairness by polarization. It is possible that ACA may be able to counteract this effect, but we believe a detailed investigation of this effect is outside the scope of this paper.
>
>
> [1] Saleem, Muniba, et al. "When and how negative news coverage empowers collective action in minorities." Communication Research 48.2 (2021): 291-316.
>
> [2] Begeny, Christopher T., et al. "The power of the ingroup for promoting collective action: How distinctive treatment from fellow minority members motivates collective action." Journal of Experimental Social Psychology 101 (2022): 104346.

---

> > ### Comment · Reviewer_YvuG · 2025-11-24
> > **Thanks for your response**
> >
> > Thanks for your response. I appreciate the clarification. In my perspective, I am still not sure whether it is practical that a substantial (e.g., 30\%) part of the minority group can collaborate with each other in full transparency and select $M$ candidates to flip their labels. I think the paper has merit but this fact makes it marginal. Currently I will keep my score.

---

> > > ### Author Response · Authors · 2025-11-25
> > >
> > > We thank the reviewer for emphasizing their concern. We would like to clarify two points.
> > >
> > > 1.  **The choice of 30%**: We highlight that the choice of 30% is somewhat arbitrary, in the sense that we highlighted this number because, according to Fig. 3, all datasets maximize their fairness benefit at 30% of the collective size. Fig. 3 shows that fairness improves relatively linearly with the collective size, meaning that there is no collective size under which ACA has no effect. Even when the size is just 10% or 20%, ACA *still works to improve fairness*. Our experiments show that ACA is always effective to some degree, and not useless for small collectives. It would of course be more desirable if we could show the same maximal benefit at 10% and less desirable if it took 40%. However, we don’t think the number 30% itself is that important to consider.
> > >
> > > 2.  **The reality of large minority participation**: There are many examples of collective action being organized and successful. In particular, there are real world collectives where minority participation rate reaches 30% and beyond. For example, in the 2020 US elections, Asian youth, facing discrimination, participated much more in activism, with 70% actively discussing these issues with their peers and 30% marching and demonstrating ([https://circle.tufts.edu/latest-research/driven-key-issues-asian-youth-increased-their-political-participation](https://circle.tufts.edu/latest-research/driven-key-issues-asian-youth-increased-their-political-participation)). In addition, ACA can also be found in many cases of digital or ML platforms, as listed in https://github.com/respML/collAction [3].
> > >
> > >
> > > The questions of self organization and coordination of ACA is a fascinating multi-disciplinary question of sociology, economy, politics and computer science. In this paper we focused on the computer science part to start discussing user-side algorithmic solutions for fairness with the hope to inspire future work to find more efficient methods and use cases.
> > >
> > >
> > >
> > > [3] Sigg, Dorothee, Moritz Hardt, and Celestine Mendler-Dünner. "Decline now: A combinatorial model for algorithmic collective action." Proceedings of the 2025 CHI Conference on Human Factors in Computing Systems. 2025.

---

> > > > ### Comment · Reviewer_YvuG · 2025-11-27
> > > >
> > > > I have no problem about the number $30\%$ being arbitrary and I understand minority group can collaborate. However, I suspect they are not able to be transparent to each other and select M representatives (e.g., you can discuss issues with your friends and your friends then discuss similar points to their friends, but the whole social graph does not share every information together, especially when you consider large demographic groups such as "Asian", "Hispanic", "African American").
> > > >
> > > > Nonetheless, I thank the authors for the comprehensive response, and I change contribution to 3. Regarding the overall rating, I still think 4 represents my overall assessment.

---

### Official Review · Reviewer_x7Ca · 2025-10-20

**Soundness:** 2
**Presentation:** 3
**Contribution:** 2
**Rating:** 6
**Confidence:** 3

**Summary:**

This paper introduces a user-driven approach (shift the focus from the firm side to the user side) to improve fairness without needing any cooperation from companies. The idea is that a coordinated group of minority users can work together to relabel their data, assigning the label they would likely receive if they were in the majority group, thereby making the model fairer when trained on such data.

**Strengths:**

- This paper frames the issue of fairness from a fresh and thought-provoking perspective. Rather than relying on traditional firm-side interventions, it shifts the focus to the user side, exploring how individuals from minority groups can collectively influence fairness without the company’s involvement. This user-driven framing feels both novel and insightful.

- The proposed strategy draws on the concept of counterfactual fairness, asking what the predicted outcome would be if an individual’s sensitive attribute were changed to that of another group. This integration of counterfactual reasoning into a user-led framework is innovative. To make this idea practical, the authors propose three ways to approximate these counterfactual labels, which makes the whole framework much more practical.

**Weaknesses:**

- The paper defines the minority and majority groups based on a binary sensitive attribute (for example, gender). However, in many real-world cases, bias may arise from other factors such as age, or even from interactions among multiple attributes (e.g., age + race + gender). Since the proposed method heavily relies on the definition of the minority group, it would be important to discuss how such groups should be identified or defined in practice.
- The three relabeling methods appear quite heuristic and may not accurately reflect the true counterfactual distribution $P(y \mid x_{A←0})$. For example, under the KNN-based approaches, if the feature space is high-dimensional, how can a simple Euclidean distance is able to capture meaningful counterfactual similarity.
- The model needs to be retrained after relabeling, which introduces additional computational costs, especially in streaming or continuously updated systems where retraining occurs in each round. In many firm-side applications, models are deployed offline and are not frequently updated. Moreover, when the minority group is very small, flipping only a few labels might have little practical impact. This suggests that the effectiveness of ACA in practice depends heavily on the platform’s architecture and feedback cycle, and this point should also be discussed.
- Trust among individuals could also be a major challenge. Coordinating collective action requires mutual confidence and shared motivation, which may not always exist in decentralized or large-scale user communities.
- My another concern is related about the method's assumption. Can these assumptions be reasonably held or implemented in practice?

**Questions:**

- In practice, how should the collective determine which users should relabel their data? What criteria or heuristics can ensure that the selected users meaningfully contribute to improving fairness rather than introducing additional noise?
- What does the size of the minority group need to be for the relabeling strategy to actually have a noticeable effect on fairness? Is there any rule of thumb or evidence for the minimum participation needed?

---

> ### Author Response · Authors · 2025-11-21
>
> We thank the reviewer for the thoughtful and constructive comments. We appreciate the positive assessment of our framing and methods.
>
> 1.  **Definition of a minority**: Although we assume a binary group membership, the ideas in the paper can be generalized to multiple groups. In this work, we have stuck to binary fairness with just two groups as common definitions of fairness (EqOd, SP, etc) in the algorithmic fairness literature (as well as most baseline algorithms) work in this regime.
>
> 2.  **Accuracy of methods**: Capturing the true counterfactual distribution is a difficult problem, and it is not clear if it is feasible with a finite amount of data or without further knowledge of the underlying causal graph. Our heuristic methods address simple causal models and don’t capture all possible cases, as we describe in lines 364-371: “relabeling according to the counterfactual implicitly assumes that the label is determined by the same features across the majority and minority, but this assumption is not valid under certain distribution shift between the group”. Further improving the use of causality is an exciting direction for future work. And indeed, KNN may not be as performant in high-dimension. This is one of the reasons that motivates us to use representation learning before applying the KNN methods (lines 407-427). Figure 8 shows that using FRL before RB-dist improves the Pareto front.
>
> 3. **Retraining**: We highlight that the firm trains the model only once after the collective relabels their data. So, there is no additional computational cost in the model training. This is similar to other ACA works [1,2,3], where the collective performs the label flipping, then contributes their data to the firm, who then trains the model. A central question in this line of work is how large the collective needs to be and how many labels need to be flipped, in order to be effective and this is precisely what we show in Figure 3, 4, 10, 11, and 12.
>
> 4. **Collective motivation**: While realistic incentives depend on the specific case, it is an interesting sociological question on when a minority can work together. Several works [4,5] show that indeed a minority is able to organize collective action when it faces discrimination. We talk about those references in lines 51-52 and in lines 480-482. We are happy to refer to more works in social science if the reviewer has further citations, however we think this is an interesting follow-up research direction to be done with experts in social science and beyond the scope of this paper.
>
> 5.  **Method’s assumptions**: We kindly ask the reviewer to specify which assumption they are concerned about so we can answer properly.
>
> 6.  **How should the collective determine which users should relabel their data**: The methods we design in section 3 are designed to choose from the collective the data points whose labels should be flipped. Each RB method provides a score for all collective members and we propose to choose those members with the highest score. The question about whether this flip helped fairness and is not just label noise is interesting and is also a central question we study in this paper. We measure this by simultaneously measuring both the error (which increases with label noise) and unfairness (which improves when the “right” labels are flipped). As our pareto plots (Figure 5, 7, 8, 13,14,15) show, our method maintains a desirable balance between fairness and error. Intuitively, this idea is based on the counterfactual assumption, and as our results show, is more effective than other baselines.
>
> 7.  **Size of the collective**: In Figures 3, 11, we plot the best EqOD that a collective can achieve at different sizes (relative to the whole minority). These figures show a consistent improvement of fairness the bigger the collective is. This improvement saturates once the collective reaches a critical size beyond which it cannot improve fairness further, as we discuss in section 5.
>
>
>
> [1] Karan, Aditya, et al. "Algorithmic Collective Action with Two Collectives." Proceedings of the 2025 ACM Conference on Fairness, Accountability, and Transparency. 2025.
>
> [2] Ben-Dov, Omri, et al. "The Role of Learning Algorithms in Collective Action." International Conference on Machine Learning. PMLR, 2024.
>
> [3] Gauthier, Etienne, Francis Bach, and Michael I. Jordan. "Statistical Collusion by Collectives on Learning Platforms." Forty-second International Conference on Machine Learning. 2025.
>
> [4] Saleem, Muniba, et al. "When and how negative news coverage empowers collective action in minorities." Communication Research 48.2 (2021): 291-316.
>
> [5] Begeny, Christopher T., et al. "The power of the ingroup for promoting collective action: How distinctive treatment from fellow minority members motivates collective action." Journal of Experimental Social Psychology 101 (2022): 104346.

---

> > ### Comment · Reviewer_x7Ca · 2025-11-24
> >
> > Thank the author for their response. The method's assumption I asked about is related to the ACA framework, for example, that assumes that minority users can coordinate and act collectively, how can this be addressed in practice?

---

> > > ### Author Response · Authors · 2025-11-24
> > >
> > > We thank the reviewer for the response and agree in general, that we need more studies on how to organize ACA in the real world. We also think that answering this specific question requires active participation from social workers, social science and political science researchers along with computer scientists. Nevertheless, we provide our analysis on whether the assumptions behind ACA are realistic.
> > > We propose to divide the assumptions related to ACA to three parts.
> > > First, whether there are scenarios where collective action is possible and useful. We address this part in Appendix A, where we analyze several scenarios where collective action, in the form proposed in the paper, is possible and can improve fairness.
> > > Second, how likely it is for a minority to conduct collective action. This is a sociological question we discuss in lines 51-53 and 480-482 and provide sociology papers that study the formation of collective action within minorities facing discrimination(Saleem et al., 2021; Begeny et al., 2022).
> > > Third is how a collective action is organized in practice. Collective action in general is a common tool to give power to the “weak”. A widely used type of collective action for example is workers unions. Collective action can also be found in many cases of digital or ML platforms, as listed in https://github.com/respML/collAction [6]. It could be argued that minorities can more easily organize collective action, through a community center, social media or group chats.
> > >
> > > [6] Sigg, Dorothee, Moritz Hardt, and Celestine Mendler-Dünner. "Decline now: A combinatorial model for algorithmic collective action." Proceedings of the 2025 CHI Conference on Human Factors in Computing Systems. 2025.

---

> > > > ### Comment · Reviewer_x7Ca · 2025-11-27
> > > >
> > > > Thank the authors for the further response. I keep my current positive rating.

---

### Official Review · Reviewer_cBnd · 2025-10-30

**Soundness:** 3
**Presentation:** 3
**Contribution:** 3
**Rating:** 6
**Confidence:** 4

**Summary:**

This paper proposes a user-side approach to fairness, where minority users coordinate to relabel their own data to reduce bias in models trained by firms, without requiring access to the training pipeline. The authors formalize this as Algorithmic Collective Action (ACA), link it to counterfactual fairness, and propose three simple, model-agnostic methods (RB-prob, RB-label, RB-dist) to approximate counterfactual relabeling. Experiments on tabular, image, and text datasets show clear fairness gains with minimal accuracy loss. The paper also analyzes theoretical limits of minority-only ACA and provides insight into how representation learning can improve fairness outcomes.

**Strengths:**

- Overall the paper is well-written and easy to follow.
- Shifts fairness work from firms to users is an interesting idea. The proposed algorithms are intuitive, easy to implement, and computationally efficient, making them appealing for real-world use by non-institutional actors.
- The experiment covers multiple datasets with analyses of Pareto frontiers, label-flip efficiency, and sensitivity to limited majority information.
- Analysis is good to have for understanding when and why minority-only action can or can’t achieve perfect fairness.

**Weaknesses:**

1. Comparisons are limited to random flipping and a few firm-side methods. I would suggest incorporating more recent pre-processing baselines (e.g., those fair subsampling or reweighting methods from existing packages[1,2]).
2. The concept of users manually relabeling their own data (e.g., resumes, engagement labels) is theoretically interesting but may not always reflect plausible or scalable user behavior. Discussion of realistic incentives, potential harms, or game-theoretic stability would enhance the social grounding.
3. I don't quite get why the waterbird dataset is being used to evaluate fairness. Can the authors offer some explanation for that?

[1] Han, Xiaotian, et al. "FFB: A Fair Fairness Benchmark for In-Processing Group Fairness Methods." ICLR 2024
[2] Bellamy, Rachel KE, et al. "AI Fairness 360: An extensible toolkit for detecting and mitigating algorithmic bias." IBM Journal of Research and Development 63.4/5 (2019): 4-1.

**Questions:**

Please refer to the weaknesses.

---

> ### Author Response · Authors · 2025-11-21
>
> We thank the reviewer for the thoughtful comments and helpful suggestions. We are glad that the clarity, scope, and analysis were appreciated.
>
> 1.  **Fairness packages**: We thank the reviewer for bringing the FBB framework to our attention. If the reviewer has specific additional methods that they see important to evaluate our contribution we would be happy to comply. Our current pipeline is based on scikit-learn, and we use the FRL and calibrated EqOD from Holistic AI ([https://github.com/holistic-ai/holisticai](https://github.com/holistic-ai/holisticai)) and the FARE implementation from the official github repo ([https://github.com/eth-sri/fare](https://github.com/eth-sri/fare)). When we compared between Holistic AI and AIF360, we decided that the Holistic AI code would fit our pipeline better. Using FBB would require creating a new codebase and pipeline.
>
> 2.  **Realistic incentives**: In appendix A we list a few scenarios where this type of collective action can be applicable. While realistic incentives depend on the specific case, it is an interesting sociological question on when a minority can work together. Several works [1,2] show that indeed a minority is able to organize collective action when it faces discrimination. We talk about those references in lines 51-52 and in lines 480-482. If the reviewer this discussion is important for the paper, we will expand on it in the revision. Finally, we also note that the standard setting of ACA involves manipulating the data features, label, or both (Hardt et. al.).
>
> 3.  **Waterbirds dataset**: Our main motivation for using this dataset is that it is commonly used in the ML literature in the context of distribution robustness, which is a closely related problem to fairness (see discussion in [3,4]). The waterbird dataset is an image dataset that is commonly used to measure the success of distributionally robust optimization (DRO) methods and, indirectly, fairness [5,6].
>
>
> [1] Saleem, Muniba, et al. "When and how negative news coverage empowers collective action in minorities." Communication Research 48.2 (2021): 291-316.
>
> [2] Begeny, Christopher T., et al. "The power of the ingroup for promoting collective action: How distinctive treatment from fellow minority members motivates collective action." Journal of Experimental Social Psychology 101 (2022): 104346.
>
> [3] Creager, Elliot, Jörn-Henrik Jacobsen, and Richard Zemel. "Environment inference for invariant learning." International Conference on Machine Learning. PMLR, 2021.
>
> [4] Koh, Pang Wei, et al. "Wilds: A benchmark of in-the-wild distribution shifts." International conference on machine learning. PMLR, 2021.
>
> [5] Sagawa, Shiori, et al. "Distributionally Robust Neural Networks." International Conference on Learning Representations (2020).
>
> [6] Liu, Evan Z., et al. "Just train twice: Improving group robustness without training group information." International Conference on Machine Learning. PMLR, 2021.

---

> > ### Comment · Reviewer_cBnd · 2025-11-25
> >
> > Thank the authors for the rebuttal. I understand it requires extra effort to accommodate methods from another toolbox, but I still believe including some methods, at least from AIF360 (which provides sklearn-style api) would strengthen the claim. I will keep my current positive score.

---

> ### Author Response · Authors · 2025-11-26
>
> We thank the reviewer for their response. We are happy to try and include comparison with a method from AIF360. Could the reviewer please let us know which exact method they would prefer us to use?

---

### Official Review · Reviewer_qA4e · 2025-10-31

**Soundness:** 3
**Presentation:** 3
**Contribution:** 2
**Rating:** 2
**Confidence:** 4

**Summary:**

This paper proposes the idea of using algorithmic collective action (a framework proposed by Hardt et al. ICML 2023) to approach the problem of group/demographic unfairness in ML models. The paper aptly argues that fairness interventions in research and in practice have been viewed primarily from the firm side, but they pose a different question: what if users from the "minority" group use their “collective action” power to input data to the ML algorithm in a way that improves group fairness metrics (like statistical parity or equalized odds) even if the ML algorithm/firm doesn’t do any interventions itself (which it does not have the incentive to do due to accuracy loss)? The paper shows through numerical experiments on a number of datasets that this specific form of data pre-processing (attained through label flipping) can be effective in making algorithms more fair.

**Strengths:**

+ The underlying idea holds merit, and is a nice, complementary approach to how fairness has been viewed primarily from firm-side.
+ The experiments also support that with the right “label flipping”, we can improve fairness.

**Weaknesses:**

While the idea is interesting, I believe there are a number of technical, practical/logical issues with it, as detailed below, as well as some lack of clarity on contributions.

- One of the main issues with ignoring the firm-side perspective is that the firm, after all, is after predictive accuracy. If the firm notices that some users are sending data to it that is decreasing its accuracy on the “majority” group (which label flipping by the "minority" group will likely do), then the firm will catch on. In other words, yes, this method does not require firm side participation, but the firm can undo the effects if it is decreasing its utility (which it likely will). This has a parallel with the idea of search engine optimization (SEO); in absence of good guardrails, it doesn’t take a lot to game search engines (a small “collective” could alter the algorithm’s recommendation sorting/output to their benefit); search engines know this well, so they have methods in place to counteract SEO. In a similar vein, firms could counteract this proposed type of collective action.

- Another main source of concern with this proposed approach is in the assumption that users cannot manipulate their features, but they can change their labels. First, why would the users not be able to change their features? If they are applying for a job, they could 100% lie on their resumes about their qualifications. Second, why would the algorithm take self-reported, flipped labels, and accept them, without verifying them? This is also an odd and hard to justify choice. It is also worth noting that there is an ever-growing literature on “strategic machine learning”: users can manipulate the data they send to algorithmic systems, in order to improve the response they get. It is the standard in this literature to assume that the only thing users can alter when reporting to the mechanism is their dataset, and there is plenty of support around why this is the only natural assumption to make. Any changes to the true label/qualification state should come from genuine improvement (not flipped label reports). In the application areas mentioned and the type of datasets used, there is a distinction between the notion of true qualification and labels assigned by an algorithm. Users can only change their true qualifications; algorithms/firms decide what label to assign to each data point, even for training; i.e., labels are not self-reported. With this view, it is hard to reconcile the assumptions (in e.g. lines 138-139, and elsewhere) that users can change their labels (and can’t change their features) with practical settings.

- On a technical note, the model seems to assume (line 95) that data for the majority and minority group is drawn i.i.d. . But then what is really distinguishing a majority group from a minority group? Also, to be clear, it would be appropriate to distinguish between “disadvantaged” and “advantaged” groups – this is the terminology that would refer to one group having lower qualification rates than others, and is the groups we want to protect through fairness interventions. The term “majority” vs. “minority” refers to data size on groups (this is not what we measure fairness against). The two notions may align (for instance, when using race as the protected attribute in the Adult dataset), but not always (for instance, when using race as the protected attribute in the COMPAS dataset). Also, on a deeper issue, if there is no relation between the feature distributions of the majority/minority group (where I’m using this paper’s terminology) and their label distributions, then why would the signal erasure strategy be relevant?

- In light of the above, the contribution of the paper seems limited. The paper, at top of Section 3, mentions rightly acknowledges that “the theory of signal erasure has been studied before”. On a related note, the claim of “we present the first practical algorithm for signal erasure” was unclear to me. The original ACA paper and its follow ups do present algorithms and establish their effectiveness, so it is not clear what was wrong or impractical about the existing algorithms?

- I am also unclear on the benchmarking done against Kamiran and Calders paper (in appendix E1). What version of CND is being used for baselines? Quoting from the Kamiran and Calders  paper: “We can now reduce the discrimination in the dataset by either promoting objects in CP from class− to + or by demoting objects in CD from class + to− (− represents the class “not +)”. In order to maintain the balance between the two classes, CND will always do both a promotion and a demotion at the same time.” Given that the way labels are flipped is not only on the minority group, comparing the methods across minority group flips does not seem immediately comparable.

**Questions:**

These question are further detailed in the weakness section. Short versions are included below as a summary.

1. How can one account for firm-side undoing of the proposed ACA effort? It seems quite reasonable that firms would try to counteract these effects. Would the proposed ACA be robust to that?

2. Why is it reasonable to assume users can report flipped labels, and that the algorithm will take them as is? What stops users from misreporting features? There is a growing line of literature on strategic machine learning which makes (and motivates) contradictory assumptions to these, with the latter seeming more relevant to/reflective of practical scenarios.

3. Why is the data assumed to be i.i.d? Will this not preclude the relevance of single erasure strategies? The notions of majority vs. minority when it comes to fairness may need to be revisited or better justified.

4. Why are the algorithms/methods proposed by prior work not considered practical? What is fundamentally different with the methods proposed here?

5. How is the paper benchmarked against works like Kamiran and Calders? Also, it seems both works (as well as many other pre-processing methods) include ideas around label flipping. Why is it that this label flipping is more effective than the label flipping achievable firm-side (like by Kamiran and Calders-type works)?

---

> ### Author Response · Authors · 2025-11-21
>
> We thank the reviewer for their insightful review and the interesting questions they ask. We are happy that they find the merit of our study of user-side fairness.
>
> 1.  **Firm-side reaction**: We agree that firm-side reaction to ACA is a natural question for the broader field of ACA. To our knowledge, existing ACA works do not directly address the issue of detection or mitigation of ACA. However, several studies found that when a firm employs different learning algorithms, such as differential privacy or distributionally robust optimization, the effect of ACA can change drastically [1,2]. How these algorithms affect signal erasure in the context of fairness deserve to be studied in future work. Including this direction in the paper would steer the paper away from the original question it poses, and also goes against our initial assumption that the firm is implementing ERM in order to maximize profit.
>
> 2.  **Flipped label justification**: Before clarifying this issue further, we would like to point out that we already highlight several motivating examples and how labels can be flipped in Appendix A. We clarify two points that the reviewer brings up.
>
>
> 	**a)**  *Why is it reasonable to assume users can report flipped labels*: In our setting, we assume that a firm is training a model for the first time on data provided by users, without pre-existing knowledge. In this case, the firm may accept labels submitted by the users without being able to know whether a label was flipped. We acknowledge that real-world firms may introduce some data cleaning or verification process. We will include this assumption explicitly in section 2.1.
>
> 	**b)** *Why not modify features*: The goal of (minority) algorithmic collective action is to collectively (from the minority) change the dataset in a way such that the model trained on that dataset will be less unfair. Importantly, this allows the model to be deployed without further data manipulation at inference time. This is precisely what we want to avoid by not using feature transformation in the collective action. As discussed in Line 136-138, when a feature is transformed during training (as commonly done in a class of fairness algorithms called Fair Representation Learning), then the transformation must be applied consistently at inference time, requiring active cooperation from each minority member to transform their features. .
>
>
> 3.  **Data iid**: To clarify the setting, we are using the standard setting where data is sampled iid from a mixture distribution where the two components of the mixture are the majority (has more weight in the distribution) and the minority (smaller weight in the mixture) groups. We agree that advantaged and disadvantaged can be more appropriate terminology when discussing fairness and we will make it clearer in the text.
>
> 4.  **Practical signal erasure**: The ACA paper by Hardt et al. studies almost exclusively the signal planting strategy (Equations (1,3) in Hardt et al.), both theoretically (Theorems 1,3 in Hardt et al.) and experimentally (Section 5 in Hardt et al.). However, while the signal erasure is studied theoretically (Theorem 5 in Hardt et al.), it is not used experimentally. This is because the erasure strategy (Eq. 4 in our paper) requires complete knowledge of the true distribution, which is usually unknown and intractable. By “we present the first practical algorithm for signal erasure” we meant that no other paper attempted to make signal erasure practical before. If the reviewer is aware of other papers that perform signal erasure, we will gladly amend this claim.
>
> 5.  **Comparison with other benchmarks**: Thank you for flagging this ambiguity about the CND comparison. CND indeed flips labels from both majority and minority, which we did as well for the experiment in appendix E1. The number of flips in figure 10 refers to the total number of flips, meaning the sum of flips from the majority and from the minority. We will add this detail explicitly in E1.
>
> [1] Solanki, Rushabh, et al. "Crowding Out The Noise: Algorithmic Collective Action Under Differential Privacy." arXiv preprint arXiv:2505.05707 (2025).
>
> [2] Ben-Dov, Omri, et al. "The Role of Learning Algorithms in Collective Action." International Conference on Machine Learning. PMLR, 2024.

---

> ### Author Response · Authors · 2025-11-27
>
> Dear reviewer qA4e,
>
> Thank you for your detailed review.  As we are halfway through the discussion period, we wanted to post a quick reminder to ask you if our above reply satisfied your concerns. If you have further questions, we are happy to discuss them further.
>
> Best,
>
> The authors

---

### Author Response · Authors · 2025-11-29
**Final remarks**

We would like to thank all reviewers for their time reading our paper and their constructive reviews. We are glad that all reviewers agree that the shift in perspective from firm-side fairness to user-side fairness is a valuable idea and that the paper is clearly written and well presented.

A concern that was brought up by some of the reviewers was how realistic it is for the type of collective action discussed in the paper to happen in the real world. In our original manuscript, we decided to focus on the machine learning aspects of our contribution and not the social aspects, similarly to prior Algorithmic Collective Action (ACA) works. That is not to say the sociological aspects are not important, and we would like to address all those concerts and describe the corresponding edits we made to the paper.

1. *In what scenarios is ACA as presented in the paper possible?* In appendix A we detail the conditions to where ACA for fairness using relabeling is possible, and list several real-world scenarios where these conditions apply.
2. *How realistic is ACA?* There are many cases where people applied ACA in the real world. The work by Sigg et al. documents and analyzes some of them [1]. In the revision we explicitly note that in the introduction with the corresponding citation.
3. *Do minorities use collective action?* In the introduction and conclusion we cite empirical sociology papers that show how minorities form collective action when facing discrimination, as they are motivated by their community and the media [2,3].
4. *Can minority ACA reach a large enough scale?* Our results show that the larger the collective is, the bigger the impact on the resulting fairness, with the effect saturating at around 30% of the minority. This amount of collaboration within minorities is possible and documented. Such as for Asian youth, whose participation can go up to 70% (https://circle.tufts.edu/latest-research/driven-key-issues-asian-youth-increased-their-political-participation). In the revision we cite two additional empirical sociology papers in the intro and conclusion to address the scaling issue, that show that a centralized organization can increase minority involvement [4,5].


We hope these points resolve the concerns about the possibility of ACA in the world.

[1] Sigg, Dorothee, Moritz Hardt, and Celestine Mendler-Dünner. "Decline now: A combinatorial model for algorithmic collective action." Proceedings of the 2025 CHI Conference on Human Factors in Computing Systems. 2025.

[2] Saleem, Muniba, et al. "When and how negative news coverage empowers collective action in minorities." Communication Research 48.2 (2021): 291-316.

[3] Begeny, Christopher T., et al. "The power of the ingroup for promoting collective action: How distinctive treatment from fellow minority members motivates collective action." Journal of Experimental Social Psychology 101 (2022): 104346.

[4] McAdam, Doug. Political process and the development of black insurgency, 1930-1970. University of Chicago Press, 1999.

[5] Michelson, Melissa R. "Meeting the challenge of Latino voter mobilization." The Annals of the American Academy of Political and Social Science 601.1 (2005): 85-101.

---

### Meta-Review · Area_Chair_6DDg · 2026-01-08

**Summary:**

Across the reviews, the setting introduced in the paper was generally seen as interesting and compelling: a creative, user-side view of fairness intervention design, framed as a pre-processing strategy where minority or vulnerable groups can act collectively to influence training outcomes without changing features at classification time. My read of the paper confirms the positive reception by the reviewers. Framing data pre-processing in terms of algorithmic collective action (ACA) is particularly compelling, turning an often neglected fairness intervention approach (it seems most methods now focus on post-processing) in a meaningful strategy for stakeholders.

At the same time, the main points of friction centered on applicability and realism, and on a perceived disconnect between the compelling ACA motivation and the scope of the experiments. The numerical results rely on familiar (dare I say, overused) fairness benchmarks and dataset choices that do not always make the ACA story as compelling as the framing suggests. Taken together, these concerns suggest the work would benefit from a tighter presentation that better connects motivation to the empirical evidence, additional experiments that clearly connect with the ACA message, and from addressing missing experimental comparisons that multiple reviewers noted.

The most negative review came from qA4e (reject, disengaged), while cBnd and x7Ca were positive and maintained their scores after discussion. YvuG engaged in discussion but maintained an overall rejecting assessment.

**Reviewer Concerns:**

A recurring concern was practicality and realism of the ACA framing. qA4e argued that focusing on label relabeling while not considering that users can manipulate features may be unrealistic in settings like hiring (e.g., updating CVs/resumes). The authors briefly responded to qA4e’s questions, but qA4e did not engage further in discussion, so it is unclear whether their concerns would be resolved by revisions.

Another recurring concern was a disconnect between motivation and experiments, including dataset choices. YvuG emphasized that some experimental choices felt arbitrary. From my view, at a higher level, that the experiments did not fully match the paper’s ACA motivation. Even after discussion with the authors, YvuG maintained their overall rejecting assessment, indicating these concerns remain outstanding from their perspective.

Reviewers also raised missing benchmark comparisons, particularly to established pre-processing baselines. cBnd was positive overall but flagged the absence of key pre-processing techniques (e.g., those available through IBM’s AIF360). While the authors discussed this point, they did not provide new benchmarking results in the discussion, leaving this as an outstanding item.

Finally, x7Ca chose to keep their positive rating, suggesting their concerns were at least partially addressed.

**Reviewer Scores:**

- qA4e had the most negative view (reject) and did not participate in the discussion. The authors offered brief clarifications, but because qA4e did not re-engage, there is no direct indication that their score would increase.

- cBnd was positive and explicitly stated they would keep their current positive score after the rebuttal/discussion. Their key remaining request was additional benchmarking (notably, AIF360-style pre-processing baselines), which was not added during discussion.

- x7Ca was positive and, after engaging with the authors, explicitly stated they would keep their current positive rating.

- YvuG engaged with the authors and maintained an overall assessment consistent with rejection. In the discussion, YvuG noted a change to the “Contribution” sub-score (to 3), but kept the overall rating at 4, indicating no overall score increase based on the discussion and maintaining a rejection vote.

---

### Decision · Program_Chairs · 2026-01-26

Reject